



# Radar measurements of blowing snow off a mountain ridge

Benjamin Walter[1], Hendrik Huwald[1,2], Josué Gehring[2], Yves Bühler[1], Michael Lehning[1,2]

[1] WSL Institute for Snow and Avalanche Research SLF, Flüelastrasse 11, 7260 Davos Dorf, Switzerland
[2] School of Architecture, Civil and Environmental Engineering, École Polytechnique Fédérale de Lausanne (EPFL),
Lausanne, Switzerland

*Correspondence to*: Benjamin Walter (walter@slf.ch)

**Abstract.** Modelling and forecasting wind-driven redistribution of snow in mountainous regions with its implications on avalanche danger, mountain hydrology or flood hazard is still a challenging task often lacking in essential details. Measurements of drifting and blowing snow for improving process understanding and model validation are typically limited
to point measurements at meteorological stations, providing no information on the spatial variability of horizontal mass fluxes or even the vertically integrated mass flux. We present a promising application of a compact and low-cost radar system for measuring and characterizing larger scale (hundreds of meters) snow redistribution processes, specifically blowing snow off a mountain ridge. These measurements provide valuable information of blowing snow velocities, frequency of occurrence, travel distances and turbulence characteristics. Blowing snow velocities measured with the radar
are validated by comparison against wind velocities measured with a 3D ultrasonic anemometer. A minimal blowing snow travel distance of 60 - 120 m is reached in 10 - 20% of the time during a snow storm, depending on the strength of the storm event. The relative frequency of transport distances decreases exponentially above the minimal travel distance, with a maximum measured distance of 280 m. The travel distance is linearly correlated with the wind velocity, revealing a threshold for snow particle entrainment and transport of 6.75 m s$^{-1}$. Turbulence statistics did not allow to draw a conclusion
on whether low-level low-turbulence jets or highly turbulent gusts are more effective in transporting blowing snow over longer distances. Drone-based photogrammetry measurements of the spatial snow height distribution revealed increased snow accumulation in the lee of the ridge being the result of the measured local blowing snow conditions.

## 1 Introduction

Seasonal and permanent snow covers in mountainous regions are of economic and environmental importance worldwide and
may affect communities in a wide range of aspects: e.g. flood hazard, avalanche danger, drinking water supply, hydropower production, lowland irrigation, ecosystem function or winter tourism (e.g. Mott et al. 2018, Grünewald et al. 2018, Beniston et al. 2018). The spatial variability of a mountain snow cover is therefore of great interest for various disciplines like natural hazard assessment, hydrology, meteorology or climatology. Orographic precipitation in mountainous regions affects the snow cover variability on larger scales (mountain range scale), whereas preferential deposition (ridge scale, e.g. Lehning et
al. 2008, Comola et al. 2019) and blowing and drifting snow (slope scale, e.g. Gerber et al. 2018, Sharma et al. 2019) are




typically responsible for local snow redistribution. The first two processes are categorized as pre-depositional and the latter one as post-depositional accumulation processes. For blowing snow, the snow particles are in suspension whereas they follow parabolic ballistic paths near the surface (saltation) for drifting snow (e.g. Bagnold 1943, Walter et al. 2014). The local mass change rate $dM/dt$ ($M$ being equivalent to the Snow Water Equivalent, $SWE$) of the snowpack (Armstrong and

Brun, 2008),

$$\frac{dM}{dt} = P - \nabla D_{bs} - E_{bs} \pm E - R \tag{1}$$

depends on the precipitation rate $P$, the horizontal redistribution rate $D_{bs}$ of surface snow by wind (drifting and blowing snow), the sublimation rate of blowing snow $E_{bs}$, sublimation/evaporation (loss of mass) or condensation/deposition (gain of mass) rates $E$ at the surface, and on the runoff rate $R$ of liquid water at the bottom of the snowpack. The objective of this

study is to gain a better understanding of the horizontal redistribution of surface snow ($D_{bs}$, mass per unit length per unit time) in mountainous terrain, especially of blowing snow off mountain ridges. To date, horizontal redistribution of snow is rather poorly investigated, difficult to measure and consequently insufficiently quantified. Because sublimation rates $E_{bs}$ of blowing snow (e.g. Groot Zwaaftink et al. 2011, Sharma et al. 2018) directly depend on the mass flux and the time snow particles are in suspension, our investigations are also relevant for better estimates of $E_{bs}$.

Despite substantial advances being made in understanding and modeling the snow cover variability in mountainous regions (e.g. Gerber et al. 2018), there is still a significant lack of in-situ measurements to better understand and characterize pre- and post-depositional accumulation processes. Point measurements of drifting and blowing snow with Snow Particle Counters (SPC, Niigata, e.g. Nishimura et al. 2014), for example at meteorological stations in mountainous terrain, do not allow for general conclusions on the spatial characteristics of snow redistribution; not even in rather close vicinity of the

station. Spatially continuous measurements of blowing snow using remote sensing techniques like radar, LIDAR (Light Detection and Ranging) or Photogrammetry may thus provide valuable information for improving our understanding and modeling of drifting and blowing snow and its spatial variability both in mountainous and flat areas on the km-scale.

This study makes use of radar measurements of blowing snow particle clouds off a mountain ridge to evaluate the potential of remote sensing techniques in characterizing pre- and post-depositional accumulation processes. It is the goal to relate

measured particle cloud characteristics like velocity distribution, transport distance and direction and turbulence intensities to the prevailing wind conditions and the subsequent snow accumulation in the vicinity. Our analysis provides a first insight into the potential of radar measurements for determining blowing snow characteristics, improves our understanding of mountain ridge blowing snow events and provides a valuable data basis for validating coupled numerical weather and snowpack simulations.

The instrumentation and methods used in this study are introduced in Section 2. In Section 3, the measured blowing snow particle cloud characteristics and meteorological conditions are presented and related to each other. A summary of the results and the conclusions from this research can be found in Section 4.





## 2 Methods

A Micro Rain Radar (MRR) was set up as a part of a meteorological Snow Drift Station (SDS) on top of the Gotschnagrat
mountain ridge at 46°51.5116N 9°50.9207E (Davos-Klosters, Switzerland) at an altitude of 2,281 m a.s.l. to investigate
drifting and blowing snow. The station was part of the 'Role of Aerosols and Clouds Enhanced by Topography on Snow'
(RACLETS) campaign, which took place in February and March 2019 in the area of Davos-Klosters. The MRR is a radar
measuring the full Doppler spectrum and operating at a frequency of 24 GHz. It is manufactured by Meteorologische
Messtechnik GmbH (METEK, Germany). The MRR is originally designed as a vertically pointing radar for measuring
clouds and precipitation. In this study, the MRR was tilted by 90° pointing horizontally to measure the particle velocity
relative to the antenna direction (Doppler velocity) and the distance of blowing snow off the Gotschnagrat mountain ridge
(Fig. 1). The Doppler spectrum provides for each Doppler velocity bin the power backscattered from particles within the
specific velocity range. From this, one can measure the mean Doppler velocity $\bar{v}$ and the spectrum width $\sigma_v$, which are
defined as:

$$\bar{v} = \frac{1}{P} \int_{-v_{ny}}^{v_{ny}} v \cdot S(v) dv \tag{2}$$

$$\sigma_v^2 = \frac{1}{P} \int_{-v_{ny}}^{v_{ny}} (v - \bar{v})^2 \cdot S(v) dv \ , \tag{3}$$

where $P = \int_{-v_{ny}}^{v_{ny}} S(v) dv$ is the mean power of the spectrum and $S(v)$ is the spectral power. Note that $v$ is weighted by $S(v)$
at each Doppler velocity bin. Since the backscattered power is more sensitive to the size of the particles than their
concentration, $v$ represents the Doppler velocity weighted by the size of the particles. The Doppler spectrum represents the
distribution of particle velocities relative to the radar. In a given radar volume, particles typically move with different
velocities due to wind turbulence, so $v$ is a measure of the mean displacement of the particles relative to the radar and $\sigma_v$ is
the standard deviation of the Doppler spectrum. In the case of a horizontally pointing antenna, $\bar{v}$ and $\sigma_v$ (hereinafter referred
to as $v_{MRR}$ and $\sigma_{v,MRR}$) can be interpreted as a measure of the mean horizontal wind velocity and turbulence.

Table 1 provides a brief overview of the MRR instrument configuration used in this study (more information in Maahn and
Kollias 2012, MRR Pro Manual 2016). It is possible to set the following five MRR configuration parameters: i) The number
of range gates $N$ = 32, 64, 128 or 256 , where a range gate defines a measurement volume of a certain length in the MRR
pointing direction, the ii) range gate length $\delta r$. The maximum measurement distance $d_{max}$ is thus defined by $N \times \delta r$; iii) The
number of lines in spectrum $m$ = 32, 64, 128 or 256 controls the velocity resolution; iv) The height above sea level $H$ of the
MRR installation site. This parameter is used for assumptions to compute rain rate from spectral power. Since it is not
relevant for this study, it was set to zero. v) The averaging time $T_i > 1$ s of the so-called power spectra defining the temporal
resolution of the MRR products (MRR Pro Manual 2016).





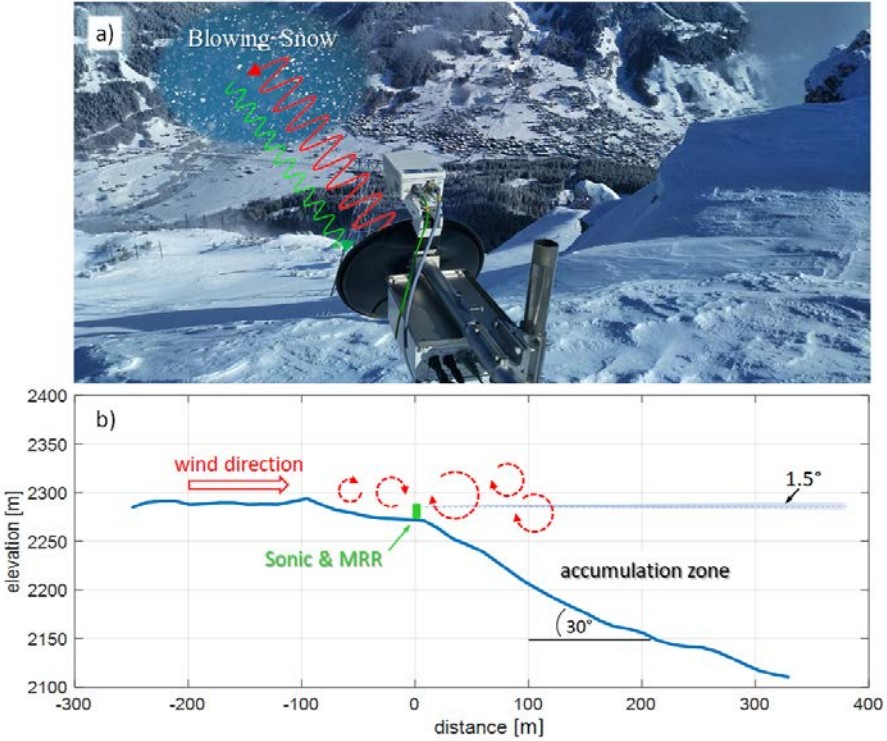

**Figure 1: a) Picture of the study site: The Micro Rain Radar (MRR) is looking horizontally from the ridge measuring the radial velocity and distance of blowing snow clouds across the valley. b) Transect of the topography in the viewing direction of the MRR**
**(aspect ratio is 1:1).**

The first range gate was removed for the analysis, since it is affected by near-field effects. The first useable range gate covers the range 20 to 40 m and the maximum measurement distance was $d_{max}$ = 1280 m for evaluation period one on 2019-03-04 (Table 1). The half power beam width of the MRR is 1.5° resulting in a beam expansion of about 1.3 m at 100 m. The

Nyquist velocity range is inverse proportional to the number of range gates $N$ (MRR Pro Manual, 2019) and was minimum for the first period with $v_{ny}$ = 24 m s$^{-1}$. The velocity resolution $\delta v$ of the MRR radial velocity $v_{MRR}$ is given by $v_{ny}$ /$m$. Because the wind direction was expected to vary depending on the general weather situation with snow potentially being blown either away or towards the MRR, the available velocity range $v_{ny}$ was set symmetrically to zero, resulting in an actual velocity range $v_{act}$ = ± $v_{ny}$ / 2 (Table 1). Velocities of $|v_{MRR}| > |v_{act}|$ resulted in aliasing but could be corrected for by applying a

dealiasing procedure based on $v_{dealiased} = v_{MRR} + n \cdot v_{ny}$, where $n$ is the dealiasing number (integer with -1 if the lower limit of the Nyquist interval is exceeded and +1 if the upper limit is exceeded). However, particle velocities $|v_{MRR}| > |v_{act}|$ were rare. Another possible source of uncertainty of the Doppler velocity is the effect of ground clutter at small range gates, where the beam is not properly formed. However, since the MRR was installed at the edge of a steep slope (30°, Fig. 1b), the effects of ground echoes on the measured Doppler velocity can be neglected. The averaging time was set to $T_i$ = 5 s for evaluation

period one and $T_i$ = 10 s for evaluation period two and three.



**Table 1: MRR parameter settings (1. - 5.) for the three different events investigated and the resulting limits (6. - 9.):**

| PARAMETER: | | 2019-03-04 | 2019-03-06/07 | 2019-03-14/15 |
|---|---|---|---|---|
| 1. Number of range gates: | $N$ | 64 | 32 | 16 |
| 2. Range gate length: | $\delta r$ [m] | 20 | 40 | 40 |
| 3. Number of lines in spectrum: | $m$ | 64 | 128 | 128 |
| 4. Height above sea level: | $H$ [m] | 0 | 0 | 0 |
| 5. Averaging time: | $T_i$ [s] | 5 | 10 | 10 |
| 6. Maximum distance: | $d_{max}$ [m] | 1280 | 1280 | 640 |
| 7. Nyquist velocity range: | $v_{ny}$ [ms-1] | 24 | 48 | 96 |
| 8. Actual velocity range: | $v_{act}$ [ms-1] | ±12 | ±24 | ±48 |
| 9. Velocity resolution: | $\delta v$ [ms-1] | 0.38 | 0.38 | 0.75 |

Among the standard products of the METEK processed data the mean MRR radial velocity $v_{MRR}$ and the spectrum width

$\sigma_{v,MRR}$ obtained for each averaging period $T_i$ are of primary interest in the subsequent analysis providing information on the

blowing snow particle cloud velocities and turbulence intensities. Furthermore, the last range gate reflecting the MRR signal

defines the blowing snow travel distances $d$ in the MRR pointing direction for each averaging period $T_i$. Finally, the radar

reflectivity $Z$, which mainly depends on the particle size, provides an indication of blowing snow particle sizes. The

determination of blowing snow particle cloud concentrations and a mass flux is not possible, since there is no quantitative

relationship between the spectral power and the particle size distribution for snow. Nevertheless, the MRR measurements

provide other interesting characteristics of blowing snow events as discussed in the following Sections.

The MRR was mounted at the edge of a few hundred meters wide flat mountain ridge transitioning into a 30° slope defining

the accumulation zone. A transect of the topography of the test site in the direction of the MRR's field of view (Fig. 2) is

shown in Fig. 1b. The MRR was oriented at an azimuth angle of 22° (clockwise with respect to north, see Fig. 2). Note that

the MRR radial velocity and turbulence characteristics determined from the MRR Doppler spectra are meant exclusively in

the direction of the field of view of the MRR. However, the wind direction $\alpha$ was typically along the MRR pointing

direction, thus the MRR radial velocity is typically close to the blowing snow absolute velocity.

Three MRR evaluation periods are in the focus of this study: 1) 2019-03-04 0400 UTC+1 – 1000 UTC+1; 2) 2019-03-06

1800 UTC+1 – 2019-03-07 0200 UTC+1 and 3) 2019-03-14 1100 UTC+1 – 1900 UTC+1. Periods one and two are the only

ones during the RACLETS campaign with strong blowing snow events in the absence of precipitation. Because the radar

signal is backscattered by all snow particles in the air, the distance of pure blowing snow events can only be obtained during

the absence of precipitation. Because both events occurred not in between two drone flights (discussed below), period three

was included in the analysis although it was a precipitation event.

At about five meters from the MRR, sensors of the SDS were mounted on a mast. The present study uses measurements of

the three wind components ($u$, $v$, $w$) and the wind direction ($\alpha$) measured with a 3D ultrasonic anemometer (R. M. Young

81000) at a height of 1.5 m above ground at a sampling frequency of 20 Hz.



Two drone flights were performed on the days 2019-03-12 and 2019-03-29 with the SenseFly eBee+ RTK fixed-wing Unmanned Aerial System (UAS) to photogrammetrically map the local snow height changes due to pre- and post-depositional snow redistribution processes in between these measurements (e.g. Schirmer et al. 2011). Photogrammetric
snow depth mapping with UAS has proven to be an accurate and reliable method to capture the spatial variability in high alpine terrain with accuracies in the range of 5 to 30 cm (Bühler et al. 2016, Harder et al. 2016, Bühler et al. 2017, Redpath et al. 2018). As a meaningful distribution of ground control points in the steep and dangerous slope was not possible, we applied integrated sensor orientation applying the UAS GNSS measurements (mean positioning accuracy: 2.5 cm). This approach proved to be valid for accurate georeferencing (Benassi et al. 2017). This is also supported by several studies we
performed for snow depth mapping applying ground control points (Bühler et al. 2018, Noetzli et al. 2019). For both flights we had a mean flight height above ground of 190 m resulting in a ground sampling distance (GSD) of about 4 cm. However, on 2019-03-12, wind gusts with high velocities up to 18 m s$^{-1}$ occurred, which led to deviations of the plane along the flight lines, resulting in a reduced overlap of the imagery. Therefore, some photogrammetric noise is present in the resulting digital surface model (DSM) reducing its accuracy (Fig. 2). No such noise is present in the data acquired on 2019-03-29, a day with
calm wind conditions. We produced two 10 cm resolution DSMs and calculated the elevation difference by subtracting them (Fig. 2).

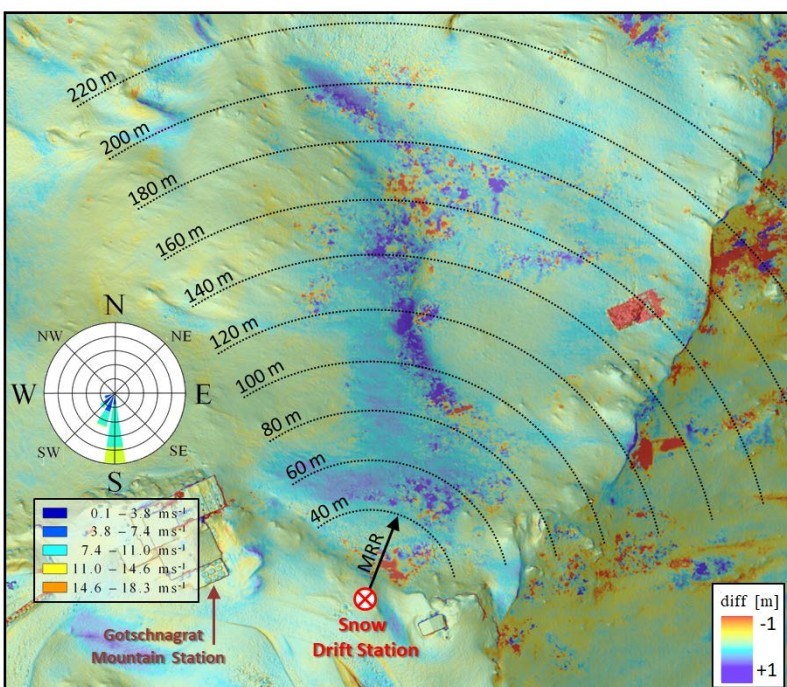

**Figure 2: Aerial view of the study domain close to the Gotschnagrat Mountain Station: Colours indicate the difference in snow height (diff) between 2019-03-29 and 2019-03-12 determined from two photogrammetry drone flights, showing areas of up to 1 m**
**of snow accumulation north of the Snow Drift Station. The horizontally aligned MRR instrument is mounted at an azimuth angle of 22° at a height of about 1 m above ground. The mean wind direction during evaluation period one (2019-03-04, 0400 UTC+1 – 1000 UTC+1) was mainly S to SW (180° - 220°) with wind velocities of up to 18.3 m s$^{-1}$.**



## 3 Results and Discussion

The spatial variation in snow height difference (diff) between 2019-03-29 and 2019-03-12 of the investigated area around
the MRR shows distinct patterns as a result of pre- and post-depositional accumulation and erosion processes (Fig. 2). The
difference shows areas of accumulation mainly north of the MRR. Strongly dark blue and dark red spotted areas of
maximum snow depth differences are an artefact from wind gusts affecting the drone flights on 2019-03-12, resulting in
erroneous photogrammetry measurements (see previous Section). Nevertheless, the snow depth map shows that snow
deposition occurred north of the SDS in between the two drone flights. A wind rose (Fig. 2) shows the wind velocity and
direction calculated from the sonic anemometer measurements for the first evaluation period (2019-03-04, 0400 UTC+1 –
1000 UTC+1).

Because the radar reflectivity $Z$ is proportional to the fourth power of the diameter for snow particles (Ryzhkov 2019), it is
mainly affected by the snow particle size and less so by the concentration as discussed before. The low reflectivity values of
the measured blowing snow clouds (not shown here), compared to the reflectivity of precipitation snowflakes, implies that
the measured blowing snow clouds were composed of rather small particles. This is consistent with other findings of drifting
and blowing snow investigations where small particle sizes of typically 50 – 500 µm were detected (Nishimura and Nemoto
2005, Gromke et al. 2014) compared to precipitation snowflakes that can have diameters of several millimetres (e.g. Gergely
et al. 2017).

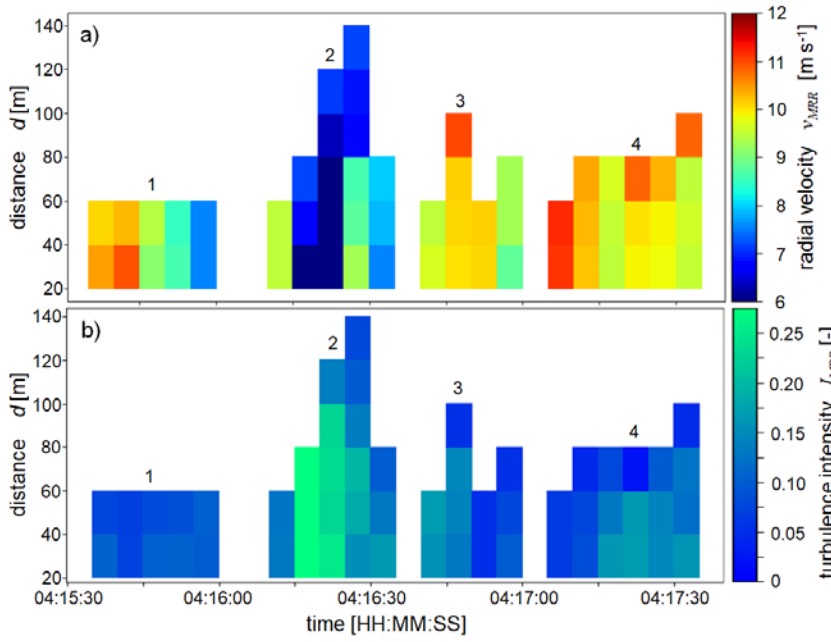


**Figure 3: a) MRR radial velocity in the azimuth direction 22° for a two-minute period containing four different blowing snow
events on 2019-03-04. b) Corresponding turbulence intensity $I$.**





### 3.1 The MRR Radial Velocity

The MRR radial velocity $v_{MRR}$ (Eq. 2) within a range gate is computed as the average of the MRR Doppler spectrum (MRR Pro Manual 2016) and is directly related to the blowing snow particle cloud velocity in the viewing direction of the MRR. In this Section we introduce the basic MRR data by means of four exemplary blowing snow events (Fig. 3) including a brief discussion and interpretation of the results as this data forms the basis for the analyses presented in the following Sections. Fig. 3a shows the MRR radial velocity $v_{MRR}$ of the four blowing snow events of different characteristics within a two-minute

time frame during the first evaluation period. The first event (No. 1) lasted for 25 s with a constant transport distance of 60 m. For the subsequent range gates (> 60 m), no snow particles were in the field of view of the MRR anymore (Fig. 1b). The assumption is that the snow was blown off the ridge horizontally by up to about 60 m before it started settling, either resulting in local accumulation or being further advected closer to the ground, and thus leaving the field of view of the MRR. Event No. 1 started with relatively high MRR radial velocities of about 10 - 11 m s$^{-1}$, while the velocities gradually

decreased to about 7-8 m s$^{-1}$ towards the end of this event. The MRR turbulence intensity $I_{MRR}$ in the direction of the MRR's field of view is defined as

$$I_{MRR} = \frac{\sigma_{v,MRR}}{v_{MRR}} , \qquad\qquad (4)$$

where the standard deviation $\sigma_{v,MRR}$ of the MRR radial velocity within each range gate is determined from the spectral width of the Doppler spectrum for each averaging period $T_i$ (Eq. 3). The turbulence intensity $I_{MRR} = 0.06 - 0.12$ of this first event

(Fig. 3b) shows low velocity fluctuations of the particle cloud, indicating a rather stable, low-level low-turbulence jet.

Blowing snow event No. 2 is different, starting with lower radial velocities of about 9 m s$^{-1}$, then suddenly dropping to about 6-7 m s$^{-1}$ during the following 10 s. Strong velocity changes are an indication for turbulent gusts which is supported by higher MRR turbulence intensities of up to $I_{MRR} = 0.27$ (Fig. 3b). The maximum turbulence intensity at the SDS measured with the sonic anemometer in the direction of the MRR during event No. 2 was $I_{Sonic} = 0.24$, thus in good agreement with the

MRR result. Generally, an overall good agreement between the turbulence intensities measured with the sonic anemometer and that of the first range gate of the MRR is found, with a mean difference of $\Delta I = mean(I_{MRR} - I_{Sonic})= 0.011$ and its standard deviation of $\sigma_{\Delta I} = 0.087$ for evaluation period one and two. The lower velocity particle cloud of event No. 2 is transported further within the field of view of the MRR compared to event No. 1, resulting in a gradually increasing transport distance starting from 60 m, increasing to 80 m, 120 m and finally to 140 m after 20 s. Events No. 3 and 4 both show rather

high radial velocities similarly to event No. 1, but also higher turbulence intensities, indicating a more turbulent flow unlike for event No. 1. The transport distances are typically 80 - 100 m for event No. 3 and 4.

Based on the above discussion of the four blowing snow events it seems that stronger turbulent fluctuations with higher turbulence intensities result in longer transport distances. That leads us to the hypothesis that not necessarily low-turbulence jets with high wind velocities but turbulent gusts with lower wind velocities may be more effective in transporting blowing

snow over longer distances on the lee side of a mountain ridge. Another explanation could be that the blowing snow cloud is

vertically more extended for turbulent gusts which increases the likelihood of snow particles being in the field of view of the MRR (Fig. 1b), whereas for low-level low-turbulence jets the particles may rather quickly settle after a certain distance, leaving the field of view of the MRR. These considerations are further discussed in Section 3.3.

### 3.2 Blowing Snow Distances

The MRR blowing snow distances $d$ for the first evaluation period are shown in Fig. 4a. Typically, a minimum distance of about 60 m is reached whereas longer distances > 100 m appear rather seldom. The distances $d$ and particle cloud radial velocities $v_{MRR}$ (Fig. 4b) may be smaller than the real absolute distances and velocities, as blowing snow from various angles (Fig. 4c), not only straight in the view direction of the MRR were detected as mentioned earlier. Nevertheless, the main wind direction was typically in overall good agreement with the view direction (202°) of the MRR (Fig. 4c), and the main interest

of this

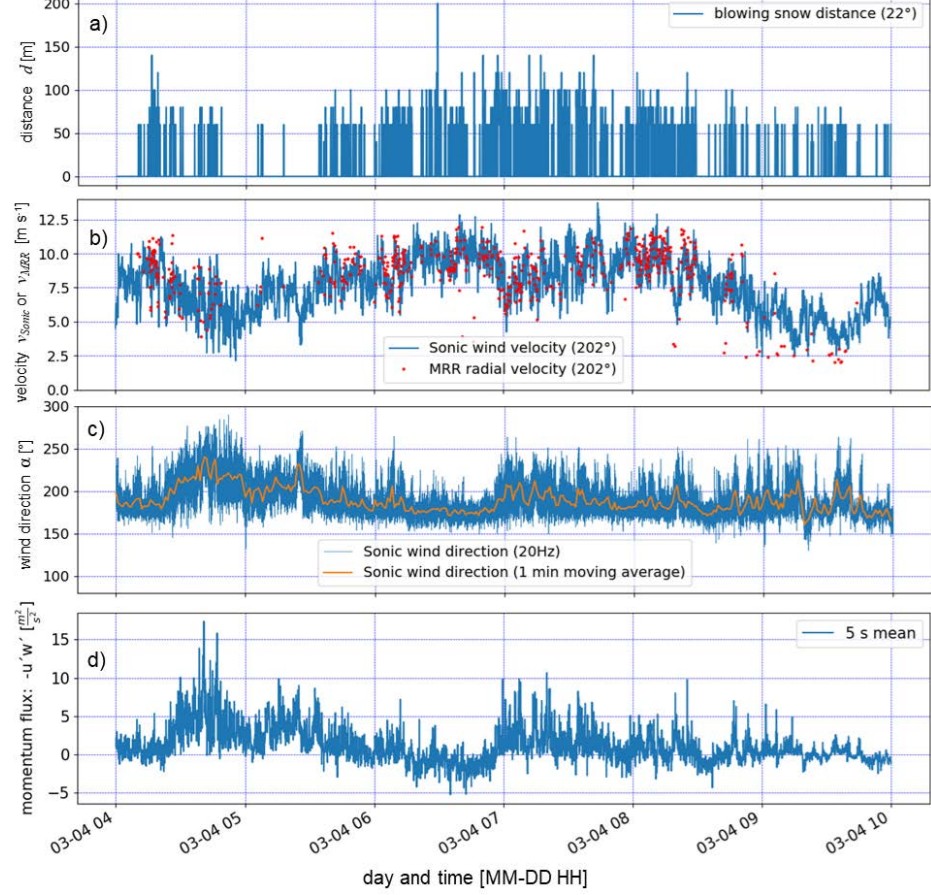

**Figure 4: a) Temporal evolution of the horizontal transport distance of all blowing snow events of evaluation period one (2019-03-04, 0400 UTC+1 – 1000 UTC+1). b) Wind velocity parallel to the MRR direction (202°) measured with a sonic anemometer compared to the close range (20 - 40m) blowing snow radial velocities measured with the MRR (see Fig. 3a). c) Wind direction**
**(mainly 180° - 220°) and d) momentum flux –u'w' calculated using the sonic anemometer data.**



study is in snow being blown off perpendicular to the Gotschnagrat mountain ridge. A comparison between the MRR radial velocities $v_{MRR}$ of the first useable range gate ($d$ = 40) and the horizontal wind velocity $v_{Sonic}$ measured with the sonic anemometer, both for the direction of 202°, is provided in Fig. 4b. A qualitatively good agreement is found despite some

outliers. Discrepancies between the two velocities may be the result of the spatial average distance of about 30 m between the first usable range gate $d$ = 40 m (with a measurement volume extending from 20 to 40 m) and the location of the sonic anemometer in combination with the slightly varying wind direction. To assess a potential dependency of the velocity difference on the wind direction, Fig. 5 shows the difference between the MRR and the sonic anemometer velocity ($v_{MRR}$ - $v_{Sonic}$) as a function of the wind direction $\alpha$ for all three evaluation periods. A slight positive trend is found with a bias of $v_{MRR}$

> $v_{Sonic}$ for wind directions $\alpha$ > 202°. Nevertheless, an overall good agreement between the MRR radial and sonic anemometer velocity is found, with a mean difference of $\Delta v = mean(v_{MRR} - v_{Sonic})$ = 0.48 m s$^{-1}$ and a standard deviation of $\sigma_{\Delta v}$ = 1.22 m s$^{-1}$. The intersection of the linear fit with the $v_{MRR}$ - $v_{Sonic}$ = 0 line for $\alpha$ = 167° (Fig. 5) suggests a stable wind direction in the vicinity of the MRR and the SDS for winds coming from that direction. This result is most likely strongly related to the local topography influencing the nearby wind field and direction, where the mountain station is located west

and another SW-NE oriented mountain ridge east of the MRR and the SDS, resulting in a rather undisturbed flow for southerly winds (Fig. 2).

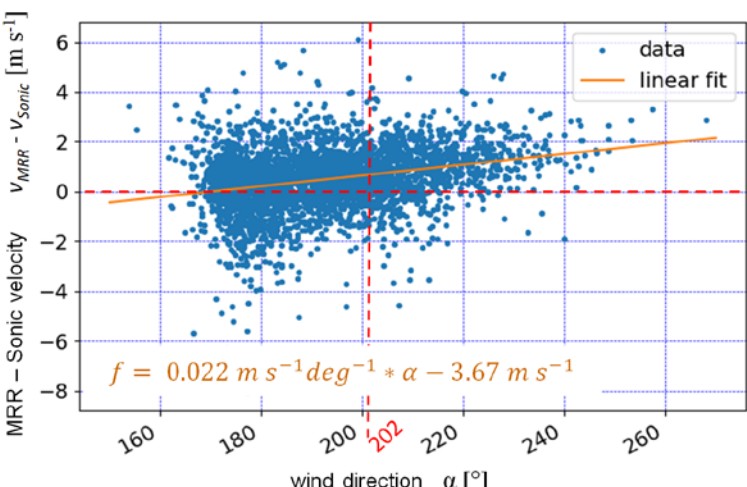

**Figure 5: Difference between MRR and sonic anemometer wind velocity in the direction 202° as a function of wind direction for all**
**three evaluation periods.**

Fig. 4d shows the momentum flux $-u`w`$ calculated from the sonic anemometer wind velocities which is generally positive for evaluation period one, indicating a downward momentum flux and an increase in wind velocity with height above the location of the sonic anemometer. However, between 0615 UTC+1 and 0700 UTC+1, the momentum flux was negative,



indicating a decreasing wind velocity with height above the sonic anemometer and the presence of a low-level jet close to the ground constantly blowing from a direction of 180° (South). During this time period, the wind velocity was highest with up to 12 - 13 m s$^{-1}$ and long blowing snow distances were reached of typically > 80 m (Fig. 4a). Furthermore, the best agreement between the sonic anemometer wind velocity and the MRR radial velocity was found for this period of stable wind conditions.

### 3.3 Blowing Snow Statistics

The relative frequency of occurrence of blowing snow transport distances from Fig. 4a is shown in Fig. 6a for the entire evaluation period one. In 80% of the time, no blowing snow was present or detected by the MRR (transport distance $d = 0$). No events were detected for a distance $d = 20$ m since this range gate cannot be used as discussed earlier. Only very few events were detected for a transport distance $d = 40$ m, although this range gate delivered continuous information on radial velocities for higher transport distances $d > 40$ m (Fig. 3a). Therefore, we expect that also for $d = 20$ m, only very few or no events would have been detected by the MRR, resulting in a gap in the frequency distribution for $0 < d < 60$ m in Fig. 6a. We hypothesize that, if the wind is strong enough and above a threshold wind speed to entrain and transport snow in suspension, a minimum transport distance of $d = 60$ m is reached, which occurred for about 10% of the total time of observation for evaluation period one (including the ´no blowing snow´ time). For distances $d > 60$ m, the relative frequency decreases exponentially with an only once observed maximum distance of $d = 200$ m. The mean sonic anemometer wind velocity was 7.3 m s$^{-1}$ during evaluation period one, which is only 6h long but sampled at a temporal resolution of 5 s resulting in 4320 samples, thus providing a good data basis for statistics.

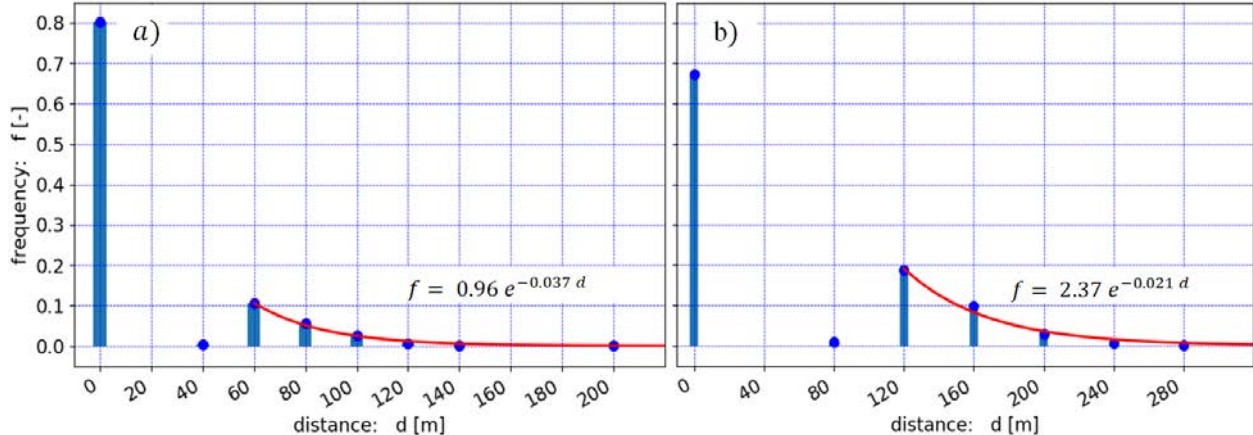

**Figure 6: Histogram of the transport distance of all blowing snow events for evaluation period a) one (Fig. 4a), and b) two, including an exponential fit for distances larger than the minimal transport distance.**





The relative frequency of occurrence of blowing snow distances for evaluation period two (2019-03-06 1800 UTC+1 –

275   2019-03-07 0200 UTC+1) is shown in Fig. 6b. The mean wind velocity of 9.1 m s$^{-1}$ during these 8h (10s sampling) was

significantly higher compared to evaluation period one (7.3 m s$^{-1}$), resulting in a larger gap before the minimal transport

distance and higher overall transport distances of up to maximum $d$ = 280 m. The higher minimal transport distance of $d$ =

120 m compared to period one might be the result of stronger gusts during the more powerful storm of evaluation period two

and the snow surface conditions and its erodibility. Despite some differences between the two distributions, both show very

similar characteristics with a gap before a minimal distance is reached and an exponential decay afterwards. Therefore, those

distributions seem to be generally valid providing a good representation of the frequency of blowing snow distances for

mountain ridges. A dependency of the minimal transport distance and the frequency distribution on the strength of the storm

event and snow cover conditions could be investigated in future more detailed studies.

To estimate a threshold wind velocity for snow particle entrainment and transport across the ridge, boxplots of the sonic

anemometer wind velocity as a function of the transport distance are provided in Fig. 7, containing data of the first two

evaluation periods. The median wind velocity increases by about 6 m s$^{-1}$ for a transport distances increasing from $d$ = 40 to $d$

= 280 m. An extrapolation of the median wind velocity to $d$ = 0 revels a potential threshold wind velocity of 6.75 m s$^{-1}$,

which is in overall good agreement with other studies (e.g. Clifton et al. 2006).

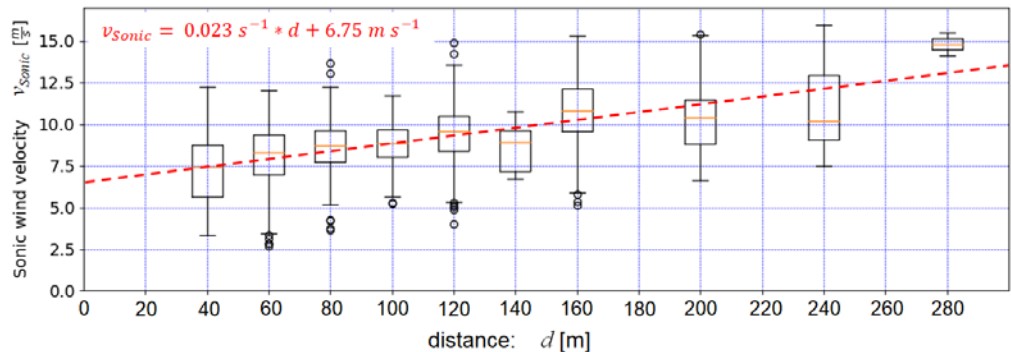

**Figure 7: Sonic anemometer wind velocity as a function of the transport distance of the blowing snow events for evaluation period one and two.**

Turbulent gusts at rather low velocities were found being potentially responsible for longer transport distances as discussed

in Section 3.1 (Fig. 3a). To investigate whether these events or low-level low-turbulence jets with high wind velocities are

more effective in transporting snow over long distances across a mountain ridge, the turbulence intensities of the last range

gate defining the blowing snow transport distance (Fig. 3b) are plotted as a function of the transport distance (box plot) in

Fig. 8. For evaluation period one (Fig. 8a) and distances $d \geq 80$ m, the median, the upper and lower quartiles, the whiskers

and the outliers all show a decreasing trend with increasing distance, indicating that low-level low-turbulence jets with high

wind velocities are more effective than highly turbulent gusts in transporting blowing snow over long distances across a





mountain ridge. Nevertheless, as mentioned earlier, highly turbulent motions still may result in a higher vertical extension of blowing snow clouds and thus in an increased likelihood of being within the field of view of the MRR (Fig. 1b) for long distances. For the stronger storm event of evaluation period two, the turbulence level was significantly higher with median intensities of $0.1 - 0.2$ ($\leq 0.5$ for period one) (Fig. 8b), supporting the latter assumption. Strong low-turbulence jets may also

result in a slight downward air flow right after the ridge and the blowing snow may quickly settle getting out of the field of view of the MRR. The turbulence statistics shown in Fig. 8 do thus not allow to draw a conclusion on whether low-level low-turbulence jets or turbulent gusts are more effective in transporting blowing snow over longer distances. Measurements with a two MRR system oriented parallel at different heights could provide a conclusion on which of the two events is more effective in transporting snow over longer distances across a mountain ridge.


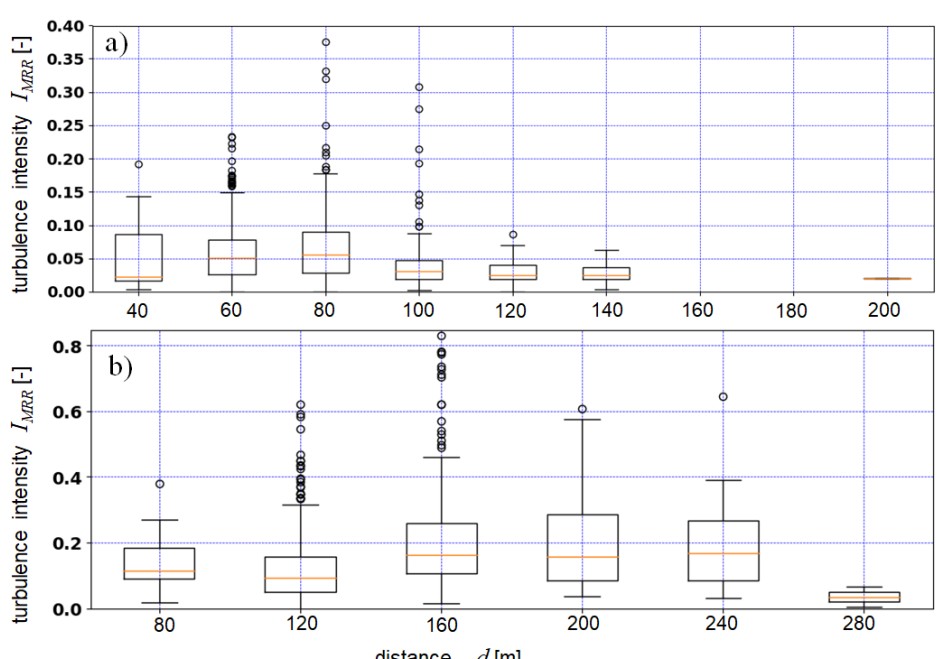

**Figure 8: Turbulence intensity determined from the MRR spectral width of the Doppler spectrum of the range gate defining the blowing snow transport distance (Fig. 3) as a function of transport distance for evaluation period a) one, and b) two.**

**3.4 Snow Height Distribution**

The increased snow accumulation north of the MRR shown in Fig. 2 is the result of a combination of preferential deposition and blowing snow, i.e. pre- and post-depositional accumulation processes. Although the pure blowing snow events analyzed in the previous sub-sections took place about a week prior to the long-term observational period, two major snow storm events were found presumably responsible for the accumulation during the 17 days between the two drone flights on 2019-



03-12 and 2019-03-29. Fig. 9a shows a comparison of the sonic anemometer wind velocity and the MRR radial velocity (similar as in Fig. 4b) for the first precipitation event on 2019-03-14 (evaluation period three). The MRR particle velocities are again in good agreement with the sonic anemometer wind velocity at similar levels of up to 8 m s$^{-1}$ as for the pure blowing snow event of evaluation period one (Section 3.1). The wind direction was aligned with the MRR view axis and quite stable from S to SW (approx. 200°) for the entire snow storm (Fig. 9b). We assume that the wind resulted in both,

preferential deposition during the precipitation event but also in snow on the ground being entrained and transported during strong gusts from the ridge to the accumulation zone (Fig. 1b, 2). This simultaneous appearance of pre- and post-depositional accumulation processes also occurred during the second snow storm on 2019-03-15, which was very similar but is not presented here. The wind rose shown in Fig. 10 summarizes the wind directions for wind velocities > 6 m s$^{-1}$, thus potentially blowing snow effective, for the 9 days period 2019-03-12 to 2019-03-21. On the latter day, the MRR and the instruments of

the SDS were dismantled. However, although the wind rose does not cover the entire period between the two drone flights, it clearly shows that the blowing snow effective wind direction was stable from S to SW at least for the first half of the time between the two drone flights.

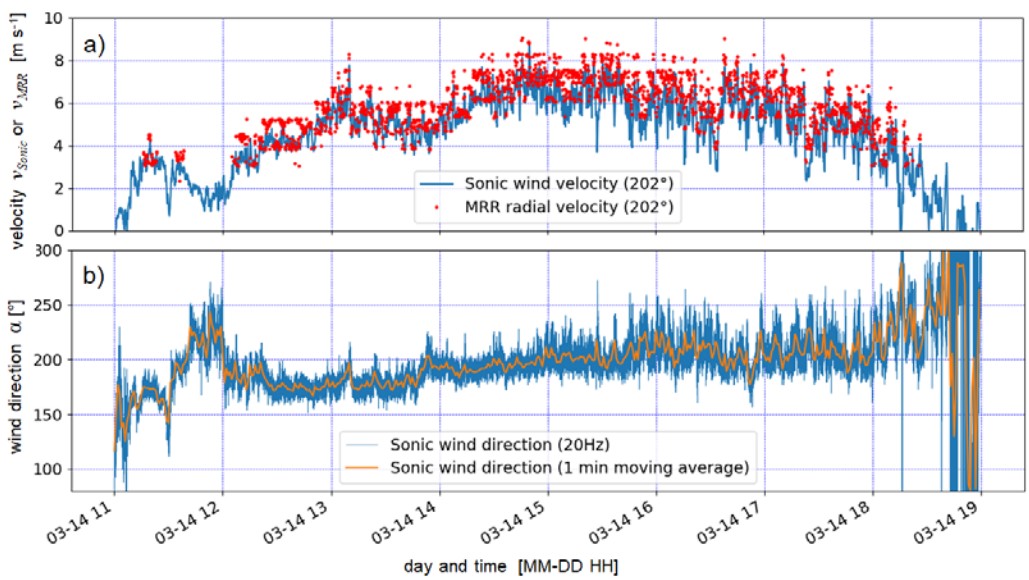

**Figure 9: Precipitation event of the evaluation period three on 2019-03-14 with strong wind from the south resulting in blowing snow and preferential deposition north of the Snow Drift Station as shown in Fig. 2. a) Sonic anemometer wind velocity and MRR radial velocity, and b) wind direction (similar as in Fig. 4b and c)**





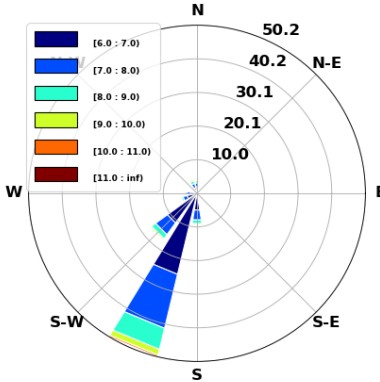

**Figure 10: Wind rose combining all major wind events with wind velocities > 6 m s⁻¹ and thus potentially blowing snow effective for the period 2019-03-12 to 2019-03-21.**

## 4 Summary and Conclusions

Our results show that radar measurements of blowing snow may deliver valuable information to improve our understanding of pre- and post-depositional snow accumulation or redistribution processes on larger scales. The Micro Rain Radar (MRR)

instrument provides characteristics of and statistics on blowing snow distances, its frequency of occurrence, particle cloud velocities and turbulence intensities. We found good agreement between the MRR blowing snow velocity and the sonic anemometer wind velocity, and that a minimal horizontal blowing snow transport distance of 60 - 120 m is reached in the lee of a mountain ridge, depending on the strength of the storm event. The relative frequency of occurrence decreases exponentially for distances longer than the minimal transport distance, with a measured maximum distance of 280 m in our

case. It was not possible to draw a conclusion on whether low-level low-turbulence jets or turbulent gusts are more effective in transporting blowing snow over longer distances in the lee of a mountain ridge. The increased snow height distribution north of the measurement location (Fig. 2) was found being the result of a combination of preferential deposition and blowing snow accumulation during at least two measured and analyzed snowstorm events.

Further investigations are required for more clarification and may incorporate measurements with a second MRR system

oriented parallel at a slightly different elevation to better resolve the local wind field and blowing snow events; particularly to capture the process of settling snow disappearing from the field of view of the upper MRR. The MRR instrument was also recently tested for measuring vertical blowing snow velocity profiles and its temporal variability in eastern Antarctica at the site S17 near the Japanese research station Syowa, where blowing snow layers can reach a vertical extend of up to 200 m (Palm et al. 2017). The next challenge for radar specialists will be finding a way to extract particle concentrations from the

radar measurements to estimate particle mass fluxes or at least its order of magnitude. Also exploring the potential of LIDAR in detecting the spatio-temporal variability of blowing snow would be worthwhile for the community interested in characterizing an better understanding pre- and post-depositional snow accumulation processes in various cold regions worldwide.



**Author contribution**

BW, HH and ML designed the experiments and BW carried them out. JG provided MRR support and YB conducted the drone flights. BW prepared the manuscript with contributions from all co-authors.

**Data availability**

The MRR and SDS data for the entire RACLETS campaign will soon be available on the ENVIDAT data repository, but are available on request by then.


**Acknowledgments**

We would like to thank the SLF workshop for supporting us with the design and construction of the Snow Drift Station. Furthermore, we would like to thank the Environmental Remote Sensing Laboratory (LTE) at EPFL, especially Alexis Berne and Alfonso Ferrone for lending the MRR and for the technical support with the instrument. Thanks also to Andreas Stoffel,

Elisabeth Hafner and Lucie Eberhard for performing the photogrammetric drone flights, David Wagner, Felix von Rütte and Beat Nett for their support with the installation of the snow drift station, and Rebecca Mott for the GIS support. J. Gehring acknowledges the financial support from the Swiss National Science Foundation (grant 200020-175700/1).

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
