# Peer review of "Radar measurements of blowing snow off a mountain ridge"

_The Cryosphere, 2019_

## Referee Comment (RC1) · Anonymous Referee #1 · 17 Jan 2020

This paper presents the use of a Micro Rain Radar (MRR) to investigate the dynamics of snow plumes forming at mountains ridges during blowing events. The MRR, pointing horizontally, was deployed at a mountain ridge above Davos in the Swiss Alps. MRR data were collected during two "pure" blowing snow events (without concurrent snowfall) and one snowfall event. Measurements provided information on the travel distance and the velocities of blowing snow in snow plumes. Snow accumulation in the lee of the ridge was also measured using drone-based photogrammetry.

The subject of this paper is very interesting for the snow community and presents novel measurements of blowing snow characteristics in mountainous terrain. To my knowledge, this is the first study that provides details measurements on the dynamics of snow plumes, which constitute the typical image of blowing snow events in alpine

terrain. So far, snow plumes have only been investigated from space (Moore, 2004) or from the air (Geerts et al., 2015). Other measurements during blowing events (fluxes, particles size and speed . . . ) are typically taken at point-scale (e.g. Naaim Bouvet et al., 2010; Nishimura et al, 2014; Aksamit and Pomeroy, 2016). These data will be very useful to evaluate blowing snow model in alpine environments. Therefore, this paper should be published in The Cryosphere. However, prior to publication, the author should clarified several points that are listed below. They are followed by more technical comments.

Comments

Abstract L 11-12: The author should mention that the number of cases studied in the paper is limited. So far, it is not clear in the abstract if the results concern one or several blowing events or even a full winter season.

Introduction: As mentioned above, this study brings novelty in the field of blowing snow studies in alpine terrain. However, so far the introduction of the paper does not reflect enough this general context and lacks an overview of the existing measurement techniques (restricted to a few sentences from L 48 to L 52 in the current version of the paper). I recommend the authors to make the distinction between measurements that are collected during blowing events and measurements that are collected before and after blowing snow events. The first kind of measurements is generally made of point measurements using Snow Particle Counters (Nishimura et al., 2014; Guyomarch et al., 2019) as already mentioned in the current introduction but also using other devices such as high-speed cameras (e.g. Aksamit and Pomeroy, 2016). The second kind of measurements usually correspond to distributed measurements such maps of snow accumulation and erosion derived from Airborne or Terrestrial Lidar Scanning or photogrammetry. This is mentioned at L 50 but without any references. These two kinds of measurements are complementary and the MRR used in this study brings a next step since it provides distributed measurements during blowing snow events. I also recommend the author to mention existing studies on snow plumes (Moore, 2004;

[Figure]

Geerts et al., 2015) and their main conclusions. In addition, the introduction is missing a paragraph on the MRR technology and its traditional use to retrieve characteristics of solid precipitation. It would be valuable for the reader to know if previous attempts have been made to study blowing snow with MRR (the authors mention such application in their conclusion L 357-358). So far, the term MRR is mentioned for the first time in Methods section. Finally, the introduction in its present form mainly contains references to papers from the Davos and Lausanne group. There are no doubt that this group has published very valuable contributions in this field but a broader perspective would certainly improve the quality of the introduction.

P 2 L 50: it is not clear here if the authors are referring here to measurements of blowing snow characteristics taken during blowing snow events or to measurements collected before and after blowing snow events. For example, when they mention radar technology, are the referring to a MRR to collect data during blowing snow events or to a ground penetrating radar to collect snow depth data before and after the event? Same for the LiDAR (see my previous comment).

P 3 L 70-75: general references on the MRR technology and its application in meteorology are missing.

P 4 Figure 1: a map of the area would be useful to better understand the experimental setting and the location of the MRR with respect to the surrounding topography. Figure 2 is not sufficient and only shows the immediate surrounding of the MRR location. The authors could also show on this map the location of the transect presented in Fig 1b.

P 4 L 97-110: this paragraph is confusing since it is the first time that the authors mention that several evaluation periods were considered in this study. The authors should re-organize this section and describe earlier the different evaluation periods. This is currently done at the end of the Methods section (P5 L 128-133). Different sets of MRR parameters settings were used for each evaluation period. The authors should explain the reasons for these different values. Did it depend on the meteorological conditions during the blowing events or the occurrence of concurrent snowfall (for evaluation period 3)?

P 7 L 159-165: it is not clear why the authors included this paragraph at the beginning of the Results section, especially since results on snow height distribution are presented later in Section 3.4.

P7 P164-166: the wind rose on Fig 2. does not bring very valuable information since it does not correspond to the period of snow depth change shown on the map. Instead, I recommend the authors to add on Fig 2 the wind rose for the full period from 12 to 21 March (date when the sonic anemometer was removed) or the wind rose only combining all major wind events during this period (as currently shown on Fig 10). In addition, it would be interesting if the author could provide at the beginning of the Results section a figure showing the two wind roses for evaluation periods one and two. This would give the reader a general overview of the wind conditions during these two events.

P 7 P 167-173: it is also not clear why the authors included this paragraph at the beginning of the Results section. A table or a figure does not support the information provided here. Since this paper constitutes the first application of a MRR to blowing snow studies in alpine terrain, I think that it would interesting to show the differences in radar reflectivity for blowing snow events with and without concurrent snowfall.

P 8 L 179: the title of this section is not appropriate since this section does not focus only on the MRR radial velocity. This section constitutes more a zoom on a specific event.

P 8 L 190-194: the MRR turbulence intensity should be defined in the Methods section at the same time as the Doppler velocity and the spectrum width.

P 10 L 227-230: the comparison between the MRR radial velocity and the horizontal velocity measured by the sonic anemometer is only carried out for the first evaluation

period. Why did the author not consider the second period as well? Is it due to the different values of range gate length between the two periods?

P 10 L 246 – P 11 L 254: The analysis of the momentum flux revealed the presence of a low-level jet close to the ground during the first evaluation period. The presence of such hump in the lowest meters has been previously reported in measurement of the wind profile at crest location (Fohn, 1980). Did the author find negative values of the momentum flux during the second evaluation period? Overall, it would be interesting to systematically carry out the same analysis for the two pure blowing snow events in Section 3.2.

P12 L 274-207. The authors mention that the average wind speed was larger during the second episode, explaining the larger transport distances. To better understand these differences of transport distance, it would be interesting to show the distributions of wind speed during the two blowing snow events and not only the average values. In a sense, Figure 7 could provide this information but the author should separate the data for the 2 blowing events. At L 279, the authors mention that the snow surface conditions and its erodibility may have been different between the two episodes. This suggests that the relationship between the transport distance and the wind speed varied between the two episodes. Separating the data on Fig. 7 would help answering this question.

P 12 L 282-283: the extrapolation of the median wind velocity to obtain a threshold velocity is rather hazardous. Indeed, the definition of the threshold velocity differs from the traditional definition of the threshold velocity for the onset of snow transport in saltation (e.g. Schmidt, 1980; Guyomarc'h and Merindol, 1998 Clifton et al., 2006). The authors should better comment on the definition of the threshold velocity and its difference with previous studies.

P 13-14: Section 3.4 presents the results on snow depth changes during the period from 12 to 29 March. This period does not correspond to the two pure blowing snow events studied in the previous sections. The authors should improve the description

of the linkage between the snow depth changes and the blowing snow characteristics derived from the MRR in Sect. 3.1 to 3.3. Indeed, so far, the MRR data in Sect. 3.4 are only used to show that the agreement is good between the MRR radial velocity and the sonic anemometer wind velocity. This was already shown in Fig 4 and 5. For example, can the author discuss similarities or differences between the transport distance from the MRR and the pattern of snow deposition in the lee of ridge? Overall, the author should better justify why showing the snow depth changes bring constructive information to this study. So far, I cannot find it and would recommend to the authors to remove this section from the paper and to focus on a mode detailed evaluation of the two blowing snow events.

P 15 L 360-364: the potential of LiDAR is not clearly defined here. Are the authors referring to Airborne Laser Scanner for measure before and after blowing snow events or vertically- (or horizontally-) pointing cloud physics Lidar for measurements during blowing snow events.

P 15: Section 4: Errors and uncertainties associated with the MRR data are not discussed in the text. It would be a very valuable addition since this paper constitutes the first investigation of the dynamics of snow plumes with a MRR and we can expect more studies to come in the future. The authors should also mention in their conclusion the potential for innovative model evaluation.

Technical Comments

Abstract L 18-19: the definition of threshold wind speed used here is questionable and a value of the threshold velocity with two decimal value may not be relevant for the abstract.

P 1 l 30: the references to Gerber et al (2018) and Sharma et al (2019) are not fully appropriate here. Indeed, the paper by Gerber et al (2018) does not study blowing and drifting snow and the paper by Sharma et al (2019) focuses on snow bedforms, which are typically below the slope scale.

P 2 L 46: the paper by Gerber et al (2018) only concerns modelling and observations of snowfall in alpine terrain. It would be valuable to add references to other studies that also consider drifting and blowing snow. See Mott et al. (2018) for a list of relevant references.

P 3 L 66-67: it would be interesting here to provide the link to the Envidat webpage that host the data collected during the campaign.

Table 1: the date for event 3 in the table differ from the date given in the text (L 129).

P 13 L 304: should it be "< 0.05 for period one"?

P 14 L 329-330: the dismantling date for the MRR and the SDS should be given in the Methods section.

References (used in this review and not present in the initial manuscript)

Aksamit, N. O., & Pomeroy, J. W. (2016). Near-surface snow particle dynamics from particle tracking velocimetry and turbulence measurements during alpine blowing snow storms. The Cryosphere, 10(6), 3043-3062.

Föhn, P. M. (1980). Snow transport over mountain crests. Journal of Glaciology, 26(94), 469-480.

Geerts, B., Pokharel, B., & Kristovich, D. A. (2015). Blowing snow as a natural glacio-genic cloud seeding mechanism. Monthly Weather Review, 143(12), 5017-5033.

Guyomarc'h, G., & Mérindol, L. (1998). Validation of an application for forecasting blowing snow. Annals of Glaciology, 26, 138-143.

Guyomarc'h, G., Bellot, H., Vionnet, V., Naaim-Bouvet, F., Déliot, Y., Fontaine, F., ... & Naaim, M. (2019). A meteorological and blowing snow data set (2000–2016) from a high-elevation alpine site (Col du Lac Blanc, France, 2720 m asl). Earth System Science Data, 11(1), 57-69.

Moore, G. W. K. (2004). Mount Everest snow plume: A case study. Geophysical research letters, 31(22).

Naaim-Bouvet, F., Bellot, H., & Naaim, M. (2010). Back analysis of drifting-snow measurements over an instrumented mountainous site. Annals of Glaciology, 51(54), 207-217.

Schmidt, R. A. (1980). Threshold wind-speeds and elastic impact in snow transport. Journal of Glaciology, 26(94), 453-467.
* * *

---

## Referee Comment (RC2) · Michele Guala (Referee) · 18 Jan 2020

Michele Guala (Referee)

mguala@umn.edu

I do value the idea to use a precipitation radar to measure the spatial extent and intensity of wind blowing snow, and I understand the difficulty to adapt an instrument to perform new types of measurements. I have a few comments, suggestions on the results and I hope the authors may decide to clarify or implement, at least some of them.

Fig. 3a) during event 2, which is the most significant one, I note opposing trends between the measured velocity and the distance. I would expect the velocity of the snow to reduce as the wind gust propagates through the accumulation slope. I interpret it as an initially concentrated jet that entrains air along its streamwise axis, lose momentum

[Figure]

as it spreads laterally causing the snow to settle on a wider area. So why is the snow velocity increasing with the distance (during some times of event 2, but also 3 and 4)? It would be interesting to correlate with the sonic to get a sense of the structure of the wind gust contributing to a blowing snow event. What is the sonic streamwise velocity time series for events 2 and 3?

Fig 4b: why the vMRR velocity occurs randomly and not necessarily at higher wind velocity. I understand sonic recording are continuous and I would expect suspended snow event to occur more systematically under strong winds.

Fig 5 : the y axis should be normalized by the sonic velocity to provide a % difference. Alternatively a scatter plot of vs versus vMRR could be provided for different ranges of directions. The figure as it is not particularly informative.

Fig 6: the exponential distribution should be assessed with log scale vertical axis. The formula are not required in my opinion as they are dimensionally questionable. Perhaps the shear velocity (from the Reynolds stress) could be introduced to normalize the distance (like a term u*ˆ2/g) ? Just a thought... May be different events could be combined under a generalized law. In general the interpretation of MRR turbulent intensity is difficult to provide and to some extent speculative. Mostly because a wind gust is a transient phenomenon and therefore any reduction in "mean" velocity with distance could be perceived as a high turbulence intensity.

Fig 7 is convincing. I am again curious about the structure of the wind gust, they might be quite coherent in both space and time to have such a lasting signature on the distance of the snow cloud. Still debated if these gusts are more like atmospheric surface layer coherent structures (see e.g. Heisel et al JFM 2018), or large sweep events that expand in the slope like a jet structure or a mixing layer.

What I suggest to the author in the next campaign, for a future paper perhaps, is to place the MRR in a flat region, such as a frozen lake and make sure that the sonic is located downstream of the MRR so that comparison in velocity could be more local, in

space and time, and over a more homogeneous topography , thus limiting as much as possible unsteady effects.

---

## Author Response (AR1)

This paper presents the use of a Micro Rain Radar (MRR) to investigate the dynamics of snow plumes forming at mountains ridges during blowing events. The MRR, pointing horizontally, was deployed at a mountain ridge above Davos in the Swiss Alps. MRR data were collected during two "pure" blowing snow events (without concurrent snowfall) and one snowfall event. Measurements provided information on the travel distance and the velocities of blowing snow in snow plumes. Snow accumulation in the lee of the ridge was also measured using drone-based photogrammetry.

The subject of this paper is very interesting for the snow community and presents novel measurements of blowing snow characteristics in mountainous terrain. To my knowledge, this is the first study that provides details measurements on the dynamics of snow plumes, which constitute the typical image of blowing snow events in alpine terrain.

***First thanks a lot for the very detailed review of our manuscript. I included basically all of your suggestions which were clear and easy to understand and which helped to significantly improve the article!***

So far, snow plumes have only been investigated from space (Moore, 2004) or from the air (Geerts et al., 2015).

***These references are included now in the Introduction Section:***

*L66: "Space born images of a huge, about 15 to 20 km long snow plume at Mount Everest have been related to local wind and weather conditions by Moore (2004). Geerts et al. (2015) used airborne radar and lidar data to show that small fractured blowing snow ice crystals may enhance snow growth in clouds."*

Other measurements during blowing events (fluxes, particles size and speed : : :) are typically taken at point-scale (e.g. Naaim Bouvet et al., 2010; Nishimura et al, 2014; Aksamit and Pomeroy, 2016).

***These references are now included in the Introduction Section:***

*L54: "... (e.g. Naaim Bouvet et al. 2010, Nishimura et al. 2014, Aksamit and Pomeroy 2016) ... "*

These data will be very useful to evaluate blowing snow model in alpine environments. Therefore, this paper should be published in The Cryosphere. However, prior to publication, the author should clarify several points that are listed below. They are followed by more technical comments.

Comments:

Abstract L 11-12: The author should mention that the number of cases studied in the paper is limited. So far, it is not clear in the abstract if the results concern one or several blowing events or even a full winter season.

***Good point, thanks!***

*L14: "Three blowing snow events are investigated, two in the absence of precipitation and one with concurrent precipitation."*

Introduction: As mentioned above, this study brings novelty in the field of blowing snow studies in alpine terrain. However, so far, the introduction of the paper does not reflect enough this general context and lacks an overview of the existing measurement techniques (restricted to a few sentences from L 48 to L 52 in the current version of the paper). I recommend the authors to make the distinction between measurements that are collected during blowing events and measurements that are collected before and after blowing snow events. The first kind of measurements is generally made of point measurements using Snow Particle Counters (Nishimura et al., 2014; Guyomarch et al., 2019) as already mentioned in the current introduction but also using other devices such as high-speed cameras (e.g. Aksamit and Pomeroy, 2016).

*With "before and after blowing snow" I guess you mean "before and after precipitation"? I think in our case it is not necessary to separate the existing literature to such two different cases. I think it is more relevant to separate between attempts of measuring and modelling spatiotemporally resolved redistribution of snow and studies based on local point measurements. Furthermore, this study focuses on blowing snow (snow particles in suspension), whereas past studies are often exclusively about drifting (saltating) snow close to the ground and therefore not necessarily relevant. Additional literature, including most of that suggested by the referees, was included in the Introduction Section with a focus on providing more details about the individual studies:*

*L54: "Naaim-Bouvet et al. (2010) used wind velocity and snow particle flux point measurements at a mountain pass to parameterize and validate a numerical model of drifting snow. Nishimura et al. (2014) measured snow particle velocities and mass fluxes using an SPC and found snow particles being about 1-2 m s$^{-1}$ slower than the wind speed below a height of 1 m. Aksamit and Pomeroy (2016) introduced an outdoor application of particle tracking velocimetry (PTV) of near-surface blowing snow investigating the complex surface flow dynamics. Despite providing valuable knowledge on process understanding, none of those studies provides spatially resolved measurements on larger scales (> 10 m)."*

*L64:" First attempts of measuring blowing snow across a mountain ridge to estimate additional snow deposition on steep lee-slopes for the local avalanche warning in Davos were presented by Föhn (1980)."*

The second kind of measurements usually correspond to distributed measurements such maps of snow accumulation and erosion derived from Airborne or Terrestrial Lidar Scanning or photogrammetry. This is mentioned at L 50 but without any references. These two kinds of measurements are complementary and the MRR used in this study brings a next step since it provides distributed measurements during blowing snow events.

*We agree that references were missing here, therefore we added those two:*

*L62: "Schirmer, M., Wirz, V., Clifton, A., and Lehning, M.: Persistence in intra-annual snow depth distribution: 1. Measurements and topographic control. Water Resources Research, 47, W09516 (16 pp.). https://doi.org/10.1029/2010WR009426, 2011.*

*Picard, G., Arnaud, L., Caneill, R., Lefebvre, E., Lamare, M.: Observation of the process of snow accumulation on the Antarctic Plateau by time lapse laser scanning, The Cryosphere, 13, 1983–1999, https://doi.org/10.5194/tc-13-1983-2019, 2019"*

*Several other important studies were also missing, so we added the following:*

*L68: "Nishimura et al. 2019 recently applied fifteen SPCs and Ultra-Sonic Anemometers on a flat field to reveal the spatio-temporal structures of blowing snow near the surface and explore the interaction with the turbulent flow structures. Several studies simulated wind-affected snow redistribution and accumulation by relating atmospheric wind fields with resulting snow deposition patterns in mountainous terrain (Dadic et al. 2010, Winstral et al. 2013, Mott et al. 2014, Vionnet et al. 2017, Gerber et al. 2017, Wang and Huang 2017)."*

I also recommend the author to mention existing studies on snow plumes (Moore, 2004; Geerts et al., 2015) and their main conclusions.
***These two studies were added as mentioned above in L66.***

In addition, the introduction is missing a paragraph on the MRR technology and its traditional use to retrieve characteristics of solid precipitation. It would be valuable for the reader to know if previous attempts have been made to study blowing snow with MRR (the authors mention such application in their conclusion L 357-358). So far, the term MRR is mentioned for the first time in Methods section.
***Thanks for this important comment! We added more information:***
*L73: "Flow structures around utility-scale 2.5 MW wind turbine have previously been measured by Hong et al. (2014) using a field Particle Imaging Velocimetry (PIV) setup with snow precipitation as the tracer particles. Their results provide significant insights into the Reynolds number similarity issues presented in wind energy applications.*
*Radar is often used for snow avalanche detection (e.g. Vriend et al. 2013) and to capture avalanche flow structures and velocities. Kneifel et al. (2011) analyzed the potential of a low-power FM-CW K-band radar (Micro Rain Radar, MRR) for snowfall observation, a method that was further improved by Maahn and Kollias (2012)."*
*L447: "The MRR instrument was also recently tested by the CRYOS group at EPFL Lausanne, Switzerland, for measuring vertical blowing snow velocity profiles and its temporal variability in eastern Antarctica at the site S17 near the Japanese research station Syowa (yet unpublished work in progress), where blowing snow layers can reach a vertical extend of up to 200 m (Palm et al. 2017)."*

Finally, the introduction in its present form mainly contains references to papers from the Davos and Lausanne group. There are no doubt that this group has published very valuable contributions in this field but a broader perspective would certainly improve the quality of the introduction.
***We agree, so we introduced more than 25 additional (mainly non-Davos/Lausanne) studies in the Introduction Section to provide a more comprehensive overview of this research field. Please see comments above.***

P 2 L 50: it is not clear here if the authors are referring here to measurements of blowing snow characteristics taken during blowing snow events or to measurements collected before and after blowing snow events. For example, when they mention radar technology, are they referring to a MRR to collect data during blowing snow events or to a ground penetrating radar to collect snow depth data before and after the event? Same for the LiDAR (see my previous comment).
***We agree that this was a bit confusing because we used a LIDAR Laser Scanner and later we mention a cloud physics LIDAR. We changed this sentence to:***
*L61: "Spatially continuous measurements using remote sensing techniques like radar, for observing blowing snow, in combination with LIDAR (Light Detection and Ranging) or Photogrammetry measurements (e.g. Schirmer et al. 2011, Picard et al. 2019), to capture the spatio-temporal snow depth variability, may thus provide valuable information for improving our understanding and modeling of drifting and blowing snow and its spatial variability."*

P 3 L 70-75: general references on the MRR technology and its application in meteorology
are missing.

*We agree, so we added some more references in the methods Section (L97 and 148):*
- *Peters, G., Fischer, B. and Andersson, T.:  Rain observations with a vertically looking Micro Rain Radar (MRR), Boreal Env. Res. 7: 353ñ362. ISSN 1239-6095, 2002.*
- *Peters, G., B. Fischer, H. Münster, M. Clemens, and Wagner A.: Profiles of raindrop size distributions as retrieved by Microrain Radars, J. Appl. Meteorol., 44, 1930–1949, 2005.*
- *Tridon, F., Van Baelen, J., Pointin, Y.: Aliasing in Micro Rain Radar data due to strong vertical winds, Geophysical Research Letters, Volume 38, Issue 2, CiteID L02804, 2011.*

P 4 Figure 1: a map of the area would be useful to better understand the experimental
setting and the location of the MRR with respect to the surrounding topography.
Figure 2 is not sufficient and only shows the immediate surrounding of the MRR location. The
authors could also show on this map the location of the transect presented in Fig 1b.
*We agree. We added a map of the surrounding topography to Fig. 2 so the reader gets an idea on the mountain ridges in the close vicinity. We also added a line in Fig. 2a indicating the location of the transect from Fig. 1 b.*

[Figure]

*Figure 2: Aerial view of the study domain close to the Gotschnagrat Mountain Station: a) Colours indicate the difference in snow height (diff) between 2019-03-29 and 2019-03-12 determined from two photogrammetry drone flights, showing areas of up to 1 m of snow accumulation north of the Snow Drift Station. The horizontally aligned MRR instrument is mounted at an azimuth angle of 22° at a height of about 1 m above ground. A wind rose indicates the wind speed and direction of all major wind events with a wind speed > 6 m s⁻¹ and thus potentially blowing snow effective for the period 2019-03-12, 1200 UTC+1 to 2019-03-21, 1200 UTC+1. b) Surrounding topography of the study site.*

P 4 L 97-110: this paragraph is confusing since it is the first time that the authors
mention that several evaluation periods were considered in this study. The authors
should re-organize this section and describe earlier the different evaluation periods.
This is currently done at the end of the Methods section (P5 L 128-133).
*Thanks. The different evaluation periods are now explained earlier in the Methods Section (L118).*

Different sets of MRR parameters settings were used for each evaluation period. The authors should explain the reasons for these different values. Did it depend on the meteorological conditions during the blowing events or the occurrence of concurrent snowfall (for evaluation period 3)?

*Very important comment, thanks a lot! It was to test different (and finding the ideal) setting(s) and not depending on the meteorological conditions. Furthermore, we think a recommendation for parameter settings would be good for those who want to perform similar experiments. We added more information here:*

*L125: "Different MRR parameter settings were tested during the RACLETS campaign to find the best setting for detecting blowing snow off mountain ridges. The most important parameters were those defining the distance and velocity resolution."*

*L160: "Providing a recommendation for an ideal MRR parameter combination is difficult, as it depends on the expected size and velocity of the blowing snow events. Based on the results of this study we recommend to start with a number of (N = 32) short (δr = 10 m) range gates resulting in a high distance resolution, a typically sufficient maximum measurement distance of 320 m and in a high Nyquist frequency of $v_{ny}$ = 48 m s$^{-1}$ ($v_{act}$ = ± 24 m s$^{-1}$). A maximum possible value of m = 256 for the number of lines in spectrum results then in a high velocity resolution of δv = 0.19 m s$^{-1}$. An averaging time of $T_i$ = 5s seems to result in a sufficient temporal resolution without producing too much data while still capturing the major flow variability."*

P 7 L 159-165: it is not clear why the authors included this paragraph at the beginning of the Results section, especially since results on snow height distribution are presented later in Section 3.4.

*Good point. We moved this paragraph to the beginning of Section 3.5.*

P7 P164-166: the wind rose on Fig 2. does not bring very valuable information since it does not correspond to the period of snow depth change shown on the map. Instead, I recommend the authors to add on Fig 2 the wind rose for the full period from 12 to 21 March (date when the sonic anemometer was removed) or the wind rose only combining all major wind events during this period (as currently shown on Fig 10). In addition, it would be interesting if the author could provide at the beginning of the Results section a figure showing the two wind roses for evaluation periods one and two. This would give the reader a general overview of the wind conditions during these two events.

*Good point. We added the wind rose from Fig. 10 to Fig. 2a as you suggested and removed Fig. 10. The wind directions for EP1, two and three are shown in (now) Fig. 5c,7c and 11b, so it is not necessary to additionally show the wind roses.*

P 7 P 167-173: it is also not clear why the authors included this paragraph at the beginning of the Results section. A table or a figure does not support the information provided here. Since this paper constitutes the first application of a MRR to blowing snow studies in alpine terrain, I think that it would interesting to show the differences in radar reflectivity for blowing snow events with and without concurrent snowfall.

*The most basic data you get from a radar is the radar reflectivity in [dB]. We wanted to begin the Results Section with some basic radar results. We agree that it was a bit curious in the previous version of the manuscript, therefore we made some changes and included a new figure (now Fig. 3), as you suggested, showing the radar reflectivity for a pure blowing snow event and one during precipitation. Thanks for this good idea!*

[Figure]

*Figure 3: MRR reflectivity for a) part of EP2 (2019-03-06 – 07) for pure blowing snow events and b) EP3 (2019-03-14) for blowing snow with concurrent snow precipitation.*

P 8 L 179: the title of this section is not appropriate since this section does not focus only on the MRR radial velocity. This section constitutes more a zoom on a specific event.
*We agree. We renamed this Section to: "Radial Velocity and Turbulence Intensity: Exemplary cases"*

P 8 L 190-194: the MRR turbulence intensity should be defined in the Methods section at the same time as the Doppler velocity and the spectrum width.
*This definition has been moved to the methods Section.*

P 10 L 227-230: the comparison between the MRR radial velocity and the horizontal velocity measured by the sonic anemometer is only carried out for the first evaluation period. Why did the author not consider the second period as well? Is it due to the different values of range gate length between the two periods?
P 10 L 246 – P 11 L 254: The analysis of the momentum flux revealed the presence of a low-level jet close to the ground during the first evaluation period. The presence of such hump in the lowest meters has been previously reported in measurement of the wind profile at crest location (Fohn, 1980). Did the author find negative values of the momentum flux during the second evaluation period? Overall, it would be interesting to systematically carry out the same analysis for the two pure blowing snow events in Section 3.2.
*We wanted to show only the details for evaluation period one (EP1) as an exemplary case (now Fig. 5), to avoid too many details and too many plots. However, we agree that the details for the second evaluation period are also relevant and interesting, therefore we included another figure (now Fig. 7) as well as additional information:*
*L 321: "Very similar results were found for EP2 (Fig. 7). Longer transport distances (Fig. 7a) were typically obtained as a result of the higher wind velocities (Fig. 7b). The wind direction (Fig. 7c) was typically quite stable although there were two periods (2100 – 2200 UTC+1 and 2300 - 2330 UTC+1) where the wind direction varied significantly. The momentum flux (Fig. 7d) was negative in about 50% of the time, indicating a higher presence of a low-level jets close to the ground compared to EP1."*

[Figure]

*Figure 7: a) Temporal evolution of the horizontal transport distance of all blowing snow events of EP2 (2019-03-06 1800 UTC+1 – 2019-03-07 0200 UTC+1). b) Wind velocity parallel to the MRR direction (202°) measured with a Sonic compared to the close range (40 - 80m) blowing snow radial velocities measured with the MRR. c) Wind direction (mainly 180° - 220°) and d) momentum flux –u'w' calculated using the Sonic data.*

**For the sake of completeness, we also included the momentum flux for the third period in Fig. 11c.**

[Figure]

*Figure 11: Precipitation event of the EP3 on 2019-03-14 with strong wind from the south resulting in blowing snow and preferential deposition north of the Snow Drift Station as shown in Fig. 2. a) Sonic wind velocity and MRR radial velocity, b) wind direction and c) momentum flux –u'w' calculated using the Sonic data (similar as in Fig. 5 and 7).*

P12 L 274-207. The authors mention that the average wind speed was larger during the second episode, explaining the larger transport distances. To better understand these differences of transport distance, it would be interesting to show the distributions of wind speed during the two blowing snow events and not only the average values. In a sense, Figure 7 could provide this information but the author should separate the data for the 2 blowing events.
***We agree, therefor we added (now) Fig. 7 (see above).***

At L 279, the authors mention that the snow surface conditions and its erodibility may have been different between the two episodes. This suggests that the relationship between the transport distance and the wind speed varied between the two episodes. Separating the data on Fig. 7 would help answering this question.
***Very good idea! Thanks! We changed this figure (now Fig. 9) as you suggested. The linear fits clearly show an increase of the "threshold velocity" for EP2. More information on this is added:***

*L352: "To estimate a threshold wind velocity (e.g. Li and Pomeroy 1997) and thus erodibility of the snow surface for particle entrainment and transport across the ridge, boxplots of the sonic anemometer wind velocity as a function of the transport distance are provided in Fig. 8. The median wind velocity increases by about 2 m s$^{-1}$ for a transport distances increasing from d = 40 - 200 m for EP1 and about 5 m s$^{-1}$ (d = 80 – 280 m) for EP2. An extrapolation of the wind velocity to d = 0 provides an estimate of a threshold velocity of 7.5 m s$^{-1}$ for EP1 and 8.8 m s$^{-1}$ for EP2, a result that is in overall good agreement with other studies (e.g. Li and Pomeroy 1997). Note: The wind velocity threshold definition for particle transport used in this study at a height of the sonic anemometer (1.5 m) is similar to that used in Li and Pomeroy (1997), who defined a threshold wind speed at 10 m above ground and is different to the traditional definition of a threshold friction velocity for particle entrainment and saltation (e.g. Schmidt 1980, Guyomarc'h and Mérindol 1998, Clifton et al. 2006, Walter et al. 2012). The fact that the estimated threshold for EP2 (Fig. 9b) is 1.3 m s$^{-1}$ higher than for EP1 (Fig. 9a) supports our previous hypothesis of different snow surface conditions with a reduced erodibility for EP2."*

P 12 L 282-283: the extrapolation of the median wind velocity to obtain a threshold velocity is rather hazardous.
***We agree, therefore we renamed it now as an "estimation of the threshold".***

Indeed, the definition of the threshold velocity differs from the traditional definition of the threshold velocity for the onset of snow transport in saltation (e.g. Schmidt, 1980; Guyomarc'h and Merindol, 1998 Clifton et al., 2006). The authors should better comment on the definition of the threshold velocity and its difference with previous studies.
***Thanks for this good comment. We included more information as shown in the comment above (L352).***

P 13-14: Section 3.4 presents the results on snow depth changes during the period from 12 to 29 March. This period does not correspond to the two pure blowing snow events studied in the previous sections. The authors should improve the description of the linkage between the snow depth changes and the blowing snow characteristics derived from the MRR in Sect. 3.1 to 3.3.
***Please see comment below: L415 cc.***

Indeed, so far, the MRR data in Sect. 3.4 are only used to show that the agreement is good between the MRR radial velocity and the sonic anemometer wind velocity. This was already shown in Fig 4 and 5.

*In the previous figures it was without precipitation, the idea was to show that the agreement also holds during precipitation. Furthermore, you suggested to show the identical data for all three events to provide a complete picture. Therefore, we decided to keep that figure and also included the momentum flux for period three (now Fig. 11c).*

For example, can the author discuss similarities or differences between the transport distance from the MRR and the pattern of snow deposition in the lee of ridge?

*We agree and added more information:*

*L 415: "Similar transport distances for the blowing snow events with concurrent precipitation (EP3) as for those without (EP1 and EP2) are assumed, based on the similarity of the wind direction and wind speed. Therefore, the increased accumulation north of the ridge up to distances of 200 m (Fig. 2a) are very likely the result of the two blowing snow events with concurrent precipitation between the two drone flights. Although the wind velocity for EP3 (Fig. 11a) are slightly smaller than for EP1 and EP2, probably resulting in smaller transport distances than shown in Fig. 5a and 7a, the snow gets likely being transported further closer to the ground outside the field of view of the MRR before it is finally deposited, which might explain increased accumulation for distances of up to d = 220 m (Fig. 2a). Although the local topography and the near ground wind velocities north of the ridge also influenced the small scale (meters) snow height distribution on the ground, the main conclusion is that an overall good agreement is found between the blowing snow direction, wind velocities, blowing snow distances and the larger scale (several tens of meters) snow accumulation pattern."*

Overall, the author should better justify why showing the snow depth changes bring constructive information to this study. So far, I cannot find it and would recommend to the authors to remove this section from the paper and to focus on a more detailed evaluation of the two blowing snow events.

*We think it is very valuable to include the snow depth variability to close the loop from Wind->blowing snow->snow redistribution->accumulation pattern. We agree that we did not properly discuss the value of including the snow height distribution. Therefore, we added more information:*

*L 442: "The presented snow height distributions together with the characterization of the blowing snow events provides a valuable data basis for validating coupled numerical weather and snowpack simulations."*

P 15 L 360-364: the potential of LiDAR is not clearly defined here. Are the authors referring to Airborne Laser Scanner for measure before and after blowing snow events or vertically- (or horizontally-) pointing cloud physics Lidar for measurements during blowing snow events.

*Thanks for this important comment. We meant the latter one:*

*L452: "Also exploring the potential of horizontally pointing cloud physics LIDAR (e.g. Mona et al. 2012) in detecting the spatio-temporal ..."*

P 15: Section 4: Errors and uncertainties associated with the MRR data are not discussed in the text. It would be a very valuable addition since this paper constitutes the first investigation of the dynamics of snow plumes with a MRR and we can expect more studies to come in the future.

*Errors of aliasing and ground clutter are described in the Methods Section (L150). More information on the uncertainty of the mean doppler velocity and the spectrum width is provided in the Methods Section:*

*L153: "Furthermore, it is difficult to quantify an uncertainty on the mean Doppler velocity $v_{MRR}$ that is a moment of a distribution, the Doppler spectrum. The measure of the Doppler velocity itself is relatively precise, i.e. depends on the precision of the clock in the radar. It is more uncertain to which extent the mean Doppler velocity is representative of the movement of the particles within a range gate. However, the main wind direction was typically well aligned with the MRR view direction and the velocity fluctuations induced by turbulence is assumed being normally distributed around the mean so that the mean Doppler velocity $v_{MRR}$ well represents the mean wind or particle velocity within a range gate."*

The authors should also mention in their conclusion the potential for innovative model evaluation. **Please see comment above. L442.**

Technical Comments

Abstract L 18-19: the definition of threshold wind speed used here is questionable and a value of the threshold velocity with two decimal value may not be relevant for the abstract.
**Thanks! This has been changed in L 19:**
*"In a first order approximation, the travel distance increases linearly with the wind velocity, allowing for an estimate of a threshold wind velocity for snow particle entrainment and transport of 7.5 – 8.8 m s$^{-1}$, most likely depending on the prevailing snow cover properties."*

P 1 l 30: the references to Gerber et al (2018) and Sharma et al (2019) are not fully appropriate here. Indeed, the paper by Gerber et al (2018) does not study blowing and drifting snow and the paper by Sharma et al (2019) focuses on snow bedforms, which are typically below the slope scale.
**This was a bit confusing as these references were referring to "slope scale" and not meant to refer to drifting and blowing snow. Slope scale can be few meters to hundreds of meters (Review Mott et al. 2018). Therefore, both papers should be OK in our opinion. Nevertheless, we added some more references:**
*L33: "Schön, P., Prokop, A., Vionnet, V., Guyomarc'h, G., Naaim-Bouvet, F., and Heiser, M.: Improving a terrain-based parameter for the assessment of snow depths with TLS data in the Col du Lac Blanc area. Cold Regions Sci. Technol. 114, 15–26. doi: 10.1016/j.coldregions.2015.02.005, 2015.*
*Shook, K., and Gray, D. M.: Small-scale spatial structure of shallow snow covers. Hydrol. Process. 10, 1283–1292, 1996."*

P 2 L 46: the paper by Gerber et al (2018) only concerns modelling and observations of snowfall in alpine terrain. It would be valuable to add references to other studies that also consider drifting and blowing snow. See Mott et al. (2018) for a list of relevant references.
**Thanks, we added two more references:**
*L49: "Guyomarc'h and Mérindol 1998, Naaim-Bouvet et al. 2010"*

P 3 L 66-67: it would be interesting here to provide the link to the Envidat webpage that host the data collected during the campaign.
**Good point! We added a reference containing the link.**
*L 93: "The data collected during the campaign including that used in this study can be found at Raclets (2019)."*
*RACLETS: Envidat data repository, https://www.envidat.ch/group/raclets-field-campaign, 2019.*

Table 1: the date for event 3 in the table differ from the date given in the text (L 129).
*We changed this.*

P 13 L 304: should it be "< 0.05 for period one"?
*We changed this.*

P 14 L 329-330: the dismantling date for the MRR and the SDS should be given in the Methods section.
*We added this to the Methods Section:*
*L123: "On 2019-03-21, the MRR and the instruments of the SDS were dismantled."*

References (used in this review and not present in the initial manuscript)
*All of the suggested references below were included in the manuscript.*

Aksamit, N. O., & Pomeroy, J. W. (2016). Near-surface snow particle dynamics from particle tracking velocimetry and turbulence measurements during alpine blowing snow storms. The Cryosphere, 10(6), 3043-3062.

Föhn, P. M. (1980). Snow transport over mountain crests. Journal of Glaciology, 26(94), 469-480.

Geerts, B., Pokharel, B., & Kristovich, D. A. (2015). Blowing snow as a natural glaciogenic cloud seeding mechanism. Monthly Weather Review, 143(12), 5017-5033.

Guyomarc'h, G., & Mérindol, L. (1998). Validation of an application for forecasting blowing snow. Annals of Glaciology, 26, 138-143.

Guyomarc'h, G., Bellot, H., Vionnet, V., Naaim-Bouvet, F., Déliot, Y., Fontaine, F., ... & Naaim, M. (2019). A meteorological and blowing snow data set (2000–2016) from a high-elevation alpine site (Col du Lac Blanc, France, 2720 m asl). Earth System Science Data, 11(1), 57-69.

rinter-friendly version

Moore, G. W. K. (2004). Mount Everest snow plume: A case study. Geophysical research letters, 31(22).

Naaim-Bouvet, F., Bellot, H., & Naaim, M. (2010). Back analysis of drifting-snow measurements over an instrumented mountainous site. Annals of Glaciology, 51(54), 207-217.

Schmidt, R. A. (1980). Threshold wind-speeds and elastic impact in snow transport. Journal of Glaciology, 26(94), 453-467.

I do value the idea to use a precipitation radar to measure the spatial extent and intensity of wind blowing snow, and I understand the difficulty to adapt an instrument to perform new types of measurements. I have a few comments, suggestions on the results and I hope the authors may decide to clarify or implement, at least some of them.
***Thanks a lot for your great ideas and suggestions. We were able to implement most of them!***

Fig. 3a) during event 2, which is the most significant one, I note opposing trends between the measured velocity and the distance. I would expect the velocity of the snow to reduce as the wind gust propagates through the accumulation slope. I interpret it as an initially concentrated jet that entrains air along its streamwise axis, lose momentum as it spreads laterally causing the snow to settle on a wider area. So why is the snow velocity increasing with the distance (during some times of event 2, but also 3 and 4)? It would be interesting to correlate with the sonic to get a sense of the structure of the wind gust contributing to a blowing snow event. What is the sonic streamwise velocity time series for events 2 and 3?

***Thanks for this good question! We added two more subplots of the sonic wind velocity and turbulence intensity to this figure and discussed it in the context of your question***
*L234: "Event No. 1 started with relatively high MRR radial velocities of about $v_{MRR}$ = 10 - 11 m s$^{-1}$, while the velocities gradually decreased to about $v_{MRR}$ = 7-8 m s$^{-1}$ towards the end of this event. The USA wind velocities (Fig. 4c) are in good agreement also decreasing to about $v_{Sonic}$ 8 m s$^{-1}$ towards the end of event No. 1. The turbulence intensity $I_{MRR}$ = 0.06 - 0.12 of this first event (Fig. 4b) shows low velocity fluctuations of the particle cloud, indicating a rather stable, low-level low-turbulence jet, which is supported by the sonic turbulence intensities (Fig. 4d). The velocity drop at the end of event No. 1 is likely the reason for the break in snow being blown off the ridge between event No. 1 and 2.*
*Blowing snow event No. 2 is different, starting with lower radial velocities of about $v_{MRR}$ = 9 m s$^{-1}$, likely being initiated by again higher wind velocities starting around 04:16:00 (Fig. 4c), then suddenly dropping to about $v_{MRR}$ = 6-7 m s$^{-1}$ during the following 10 s because of another wind velocity $v_{Sonic}$ decrease around 04:16:10 (Fig. 4c). Strong velocity changes are an indication for turbulent gusts which is supported by higher MRR turbulence intensities of up to $I_{MRR}$ = 0.27 (Fig. 4b). The maximum turbulence intensity at the SDS measured with the Sonic in the direction of the MRR during event No. 2 was $I_{Sonic}$ = 0.25 (Fig. 4d), thus in good agreement with the MRR result. However, the temporal agreement of the peak turbulence intensities is rather poor, as the peak in $I_{Sonic}$ lags the peak in $I_{MRR}$ although it should be vice versa. Nevertheless, an overall good agreement between the turbulence intensities measured with the Sonic and that of the first range gate of the MRR is found, with a mean difference of $\Delta I$ = mean($I_{MRR}$ - $I_{Sonic}$) = 0.011 and its standard deviation of $\sigma_{\Delta I}$ = 0.087 for the entire EP1 and EP2. The lower velocity particle cloud of event No. 2 is transported further within the field of view of the MRR compared to event No. 1, resulting in*

*a gradually increasing transport distance starting from 60 m, increasing to 80 m, 120 m and finally to 140 m after 20 s. Interestingly, $v_{MRR}$ is increasing with distance for event No. 2, which is counter-intuitive, as one would rather expect a decrease of the wind velocity behind the ridge. However, the highly turbulent flow with changes in the wind direction and potentially large eddies of up to 100 m is likely causing this effect of higher velocities at longer distances. Events No. 3 and 4 both show rather high radial velocities similarly to event No. 1 and supported by the Sonic wind velocities (Fig. 4c), but also slightly higher turbulence intensities, indicating a more turbulent flow unlike for event No. 1. The transport distances are about 80 - 100 m for event No. 3 and 4."*

[Figure]

*Figure 4: a) MRR radial velocity in the azimuth direction 22° for a two-minute period containing four different blowing snow events on 2019-03-04. b) Corresponding turbulence intensity I, USA c) wind velocity and d) turbulence intensity.*

Fig 4b: why the vMRR velocity occurs randomly and not necessarily at higher wind velocity. I understand sonic recording are continuous and I would expect suspended snow event to occur more systematically under strong winds.

***We agree. Generally, a trend for MRR velocities occurring at higher wind speeds is given, e.g. around 05:00 and after 08:30 (now Fig. 5) the wind velocities were small and only few MRR events were detected. The outliers, e.g. low MRR velocities around 2.5 m/s after 08:30 are most likely instrument artefacts. We added one more sentence on this in***

*L285: "Very low MRR velocities around $v_{MRR} = 2.5$ m s$^{-1}$ are either an instrument artefact because of very low blowing snow particle concentrations, or wind directions temporarily deviating significantly from the MRR field of view direction."*

Fig 5: the y axis should be normalized by the sonic velocity to provide a % difference. Alternatively, a scatter plot of vs versus vMRR could be provided for different ranges of directions. The figure as it is not particularly informative.

***Good idea, this has been done and the text been changed accordingly!***

*L290 "To assess a potential dependency of the velocity difference on the wind direction, Fig. 6 shows the relative difference between the MRR and the Sonic velocity as a function of the wind direction α for all three evaluation periods. A positive trend is found with a bias of $v_{MRR} > v_{Sonic}$ for wind directions α > 180°. Nevertheless, an overall good agreement between the MRR radial and SONIC velocity is found, with a mean difference of mean(($v_{MRR} - v_{Sonic}$) / $v_{Sonic}$) = 10% and a standard deviation of ± 20%. The intersection of the linear fit with the $v_{MRR} - v_{Sonic} = 0$ line for α = 170° (Fig. 6) suggests a stable wind direction in the vicinity of the MRR and the SDS for winds coming from that direction. This result is most likely strongly related to the local topography (Fig. 2b) influencing the nearby wind field and direction, where the mountain station is located west and another SW-NE oriented mountain ridge east of the MRR and the SDS, resulting in a rather undisturbed flow for southerly winds. "*

[Figure]

***Figure 6: Relative difference between MRR and Sonic wind velocity in the direction 202• as a function of wind direction for all three evaluation periods.***

Fig 6: the exponential distribution should be assessed with log scale vertical axis. The formula are not required in my opinion as they are dimensionally questionable.

***We totally agree, so we changed this. Thanks!***

[Figure]

*Figure 8: Histogram of the transport distance of all blowing snow events for a) EP1 (Fig. 5a), and b) EP2 (Fig. 7a), including exponential fits for distances larger than the minimal transport distance.*

Perhaps the shear velocity (from the Reynolds stress) could be introduced to normalize the distance (like a term u*^2/g) ? Just a thought... May be different events could be combined under a generalized law.

*We think for a first attempt of characterizing blowing snow off mountain ridges with a radar it is OK to keep the real distances, this makes it easier for the reader e.g. when comparing it to the snow height distribution in Fig. 2a. Furthermore, we have only 2 distributions (events). Finding a generalized law would require more blowing snow events for different conditions, e.g. wind, snow surface, etc. Therefore, we decided to keep it as it is leaving this for future studies.*

In general, the interpretation of MRR turbulent intensity is difficult to provide and to some extent speculative. Mostly because a wind gust is a transient phenomenon and therefore any reduction in "mean" velocity with distance could be perceived as a high turbulence intensity.

*We agree, therefore we added more information in*

*L114: "The definition of $I_{MRR}$ includes the assumption that within each range gate of length $\delta r$ and for each time interval $T_i$ the MRR velocity is normally distributed around the mean velocity $v_{MRR}$. This assumption is supported by the good agreement between the MRR turbulence intensity $I_{MRR}$ and the turbulence intensity $I_{Sonic}$ determined from a 3D Ultra-Sonic anemometer (Sonic) as will be shown in Section 3.2."*

Fig 7 is convincing. I am again curious about the structure of the wind gust, they might be quite coherent in both space and time to have such a lasting signature on the distance of the snow cloud. Still debated if these gusts are more like atmospheric surface layer coherent structures (see e.g. Heisel et al JFM 2018), or large sweep events that expand in the slope like a jet structure or a mixing layer.

*We changed Fig. 7 (now Fig. 9) to separate the two evaluation periods (See comment of other reviewer above). We agree that it would be great to further investigate the flow structures with a better setup. However, the goal of this study was to introduce a new method for characterizing blowing snow on larger spatial scales, which certainly leaves room for more detailed future studies building upon the here presented results.*

What I suggest to the author in the next campaign, for a future paper perhaps, is to place the MRR in a flat region, such as a frozen lake and make sure that the sonic is located downstream of the MRR so that comparison in velocity could be more local, in space and time, and over a more homogeneous topography, thus limiting as much as possible unsteady effects.

***Thanks for this valuable suggestion, something similar has already been done as mentioned:***

*L447: "The MRR instrument was also recently tested by the CRYOS group at EPFL Lausanne, Switzerland, for measuring vertical blowing snow velocity profiles and its temporal variability in eastern Antarctica at the site S17 near the Japanese research station Syowa (unpublished work in progress), where blowing snow layers can reach a vertical extend of up to 200 m (Palm et al. 2017)."*

**Radar measurements of blowing snow off a mountain ridge**

[revised manuscript text omitted]

Despite substantial advances being made in understanding and modeling theblowing snow and the resulting snow cover variability in mountainous regions (e.g. Guyomarc'h and Mérindol 1998, Naaim-Bouvet et al. 2010, Gerber et al. 2018, Mott

50 et al. 2018), there is still a significant lack of in-situ measurements to better understand and characterize pre- and post-depositional accumulation processes. Point measurements of drifting and blowing snow with Snow Particle Counters (SPC, Niigata, e.g. Nishimura et al. 2014, Guyomarc'h et al. 2019), for example at meteorological stations in mountainous terrain, do not allow for general conclusions on the spatial characteristics of snow redistribution; not even in rather close vicinity of the station. (e.g. Naaim-Bouvet et al. 2010, Nishimura et al. 2014, Aksamit and Pomeroy 2016). Naaim-Bouvet et al. (2010)

55 used point measurements of the wind velocity and snow particle flux at a mountain pass to parameterize and validate a numerical model of drifting snow. Nishimura et al. (2014) measured snow particle velocities and mass fluxes using an SPC and found snow particles being about 1-2 m s$^{-1}$ slower than the wind speed below a height of 1 m. Aksamit and Pomeroy (2016) introduced an outdoor application of particle tracking velocimetry (PTV) of near-surface blowing snow investigating the complex surface flow dynamics. Despite providing valuable knowledge on process understanding, none of those studies

60 provides spatially resolved measurements on larger scales (> 10 m).

Spatially continuous measurements of blowing snow using remote sensing techniques like radar, for observing blowing snow, in combination with LIDAR (Light Detection and Ranging) or Photogrammetry measurements (e.g. Schirmer et al. 2011, Picard et al. 2019), to capture the spatio-temporal snow depth variability, may thus provide valuable information for improving

our understanding and modeling of drifting and blowing snow and its  spatiotemporal variability. First attempts of measuring blowing snow across a mountain ridge to estimate additional snow deposition on steep lee-slopes for the local avalanche warning in Davos were presented by Föhn (1980). Space born images of a huge, about 15 to 20 km long snow plume at Mount Everest have been related to local wind and weather conditions by Moore (2004). Geerts et al. (2015) used airborne radar and lidar data to show that small fractured blowing snow ice crystals may enhance snow growth in clouds. Nishimura et al. 2019 recently applied fifteen SPCs and ultra-sonic anemometers on a flat field to reveal the spatio-temporal structures of blowing snow near the surface and explore the interaction with the turbulent flow structures. Several studies simulated wind-affected snow redistribution and accumulation by relating atmospheric wind fields with resulting snow deposition patterns in mountainous terrain (Dadic et al. 2010, Winstral et al. 2013, Mott et al. 2014, Vionnet et al. 2017, Gerber et al. 2017, Wang and Huang 2017). Flow structures around a utility-scale 2.5 MW wind turbine have previously been measured by Hong et al. (2014) using a field Particle Imaging Velocimetry (PIV) setup with snow precipitation as the tracer particles. Their results provide significant insights into the Reynolds number similarity issues presented in wind energy applications.

Radar is often used for snow avalanche detection (e.g. Vriend et al. 2013) and to capture avalanche flow structures and velocities. Kneifel et al. (2011) analyzed the potential of a low-power FM-CW K-band radar (Micro Rain Radar, MRR) for snowfall observation, a method that was further improved by Maahn and Kollias (2012). This study makes use of ground radar measurements of blowing snow particle clouds off a mountain ridge using an MRR instrument to evaluate the potential of remote sensing techniques in characterizing pre- and post-depositional accumulation processes. The goal is to relate measured particle cloud characteristics like velocity distribution, transport distance and direction and turbulence intensities to the prevailing wind conditions and the subsequent snow accumulation in the vicinity. Our analysis provides a first insight into the potential of radar measurements for determining blowing snow characteristics, improves our understanding of mountain ridge blowing snow events and provides a valuable data basis for validating coupled numerical weather and snowpack simulations.

The instrumentation and methods used in this study are introduced in Section 2. In Section 3, the measured blowing snow particle cloud characteristics  meteorological conditions and snow distributions are presented, discussed and related to each other. A summary of the results and the conclusions from this research can be found in Section 4.

**2 Methods**

A Micro Rain Radar (MRR) was set up as a part of a meteorological Snow Drift Station (SDS) on top of the Gotschnagrat mountain ridge at 46°51.5116N 9°50.9207E (Davos-Klosters, Switzerland) at an altitude of 2,281 m a.s.l. to investigate drifting and blowing snow. The station was part of the 'Role of Aerosols and Clouds Enhanced by Topography on Snow' (RACLETS) campaign, which took place in February and March 2019 in the area of Davos-Klosters. The data collected during the campaign, including that used in this study, has been made publicly available (Raclets 2019). The MRR is a radar measuring

the full Doppler spectrum and operating at a frequency of 24 GHz. It is manufactured by Meteorologische Messtechnik GmbH (METEK, Germany). The MRR is originally designed as a vertically pointing radar for measuring clouds and precipitation. (Peters et al. 2002 and 2005). In this study, the MRR was tilted by 90° pointing horizontally to measure the particle velocity relative to the antenna direction (Doppler velocity) and the distance of blowing snow off the Gotschnagrat mountain ridge

100 (Fig. 1). The Doppler spectrum provides for each Doppler velocity bin the power backscattered from particles within the specific velocity range. From this, one can determine the mean Doppler velocity $\bar{v}$ and the spectrum width $\sigma_v$, which are defined as:

$$\bar{v} = \frac{1}{P} \int_{-v_{ny}}^{v_{ny}} v \cdot S(v) dv \tag{2}$$

$$\sigma_v^2 = \frac{1}{P} \int_{-v_{ny}}^{v_{ny}} (v - \bar{v})^2 \cdot S(v) dv \ , \tag{3}$$

105 where $P = \int_{-v_{ny}}^{v_{ny}} S(v) dv$ is the mean power of the spectrum and $S(v)$ is the spectral power. Note that $v$ is weighted by $S(v)$ at each Doppler velocity bin. Since the backscattered power is more sensitive to the size of the particles than their concentration, $v$ represents the Doppler velocity weighted by the size of the particles. The Doppler spectrum represents the distribution of particle velocities relative to the radar. In a given radar volume, particles typically move with different velocities due to wind turbulence, so $v$ is a measure of the mean displacement of the particles relative to the radar and $\sigma_v$ is the standard deviation of

110 the Doppler spectrum. In the case of a horizontally pointing antenna, $\bar{v}$ and $\sigma_v$ (hereinafter referred to as $v_{MRR}$ and $\sigma_{v,MRR}$) can be interpreted as a measure of the mean horizontal wind velocity and turbulence.

The MRR turbulence intensity $I_{MRR}$ in the direction of the MRR's field of view is defined as

$$I_{MRR} = \frac{\sigma_{v,MRR}}{v_{MRR}}, \tag{4}$$

where the standard deviation $\sigma_{v,MRR}$ of the MRR radial velocity within each range gate is determined from the spectral width

115 of the Doppler spectrum for each averaging period $T_i$. The definition of $I_{MRR}$ includes the assumption that within each range gate of length $\delta r$ and for each time interval $T_i$, the MRR velocity is normally distributed around the mean velocity $v_{MRR}$. This assumption is supported by the good agreement between the MRR turbulence intensity $I_{MRR}$ and the turbulence intensity $I_{Sonic}$ determined from a 3D Ultra-Sonic anemometer (Sonic) as will be shown in Section 3.2.

Three MRR evaluation periods (EP) are in the focus of this study: 1) 2019-03-04 0400 UTC+1 – 1000 UTC+1 (EP1); 2) 2019-

120 03-06 1800 UTC+1 – 2019-03-07 0200 UTC+1 (EP2) and 3) 2019-03-14 1100 UTC+1 – 1900 UTC+1 (EP3). EP1 and EP2 are the only ones during the RACLETS campaign with strong blowing snow events in the absence of precipitation. Because the radar signal is backscattered by all snow particles in the air, the distance of pure blowing snow events can only be obtained without precipitation. Because both events occurred not in between two drone flights (discussed below), EP3 was included in the analysis, although it was a precipitation event. On 2019-03-21, the MRR and the instruments of the SDS were dismantled.

125 Different MRR parameter settings were tested during the RACLETS campaign to find the best setting for detecting blowing snow off mountain ridges. The most important parameters were those defining the distance and velocity resolution. Table 1 provides a brief overview of the MRR instrument configuration used in this study (more information in Maahn and Kollias 2012, and MRR Pro Manual 2016). It is possible to set the following five MRR configuration parameters: i) The number of range gates $N$ = 32, 64, 128 or 256, where a range gate defines a measurement volume of a certain length in the MRR pointing

130 direction, the ii) range gate length $\delta r$ (> 10m). The maximum measurement distance $d_{max}$ is thus defined by $N \times \delta r$; iii) The number of lines in spectrum $m$ = 32, 64, 128 or 256 controls the velocity resolution; iv) The height above sea level $H$ of the MRR installation site. This parameter is used for assumptions to compute rain rate from spectral power. Since it is not relevant for this study, it was set to zero. v) The averaging time $T_i$ > 1 s of the so-called power spectra defining the temporal resolution of the MRR products (MRR Pro Manual 2016).

[Figure]

135

[Figure]

**Figure 1: a) Picture of the study site: The Micro Rain Radar (MRR) is looking horizontally from the ridge measuring the radial velocity and distance of blowing snow clouds across the valley. b) Transect of the topography in the viewing direction of the MRR (aspect ratio is 1:1).**

The first range gate was removed for the analysis, since it is affected by near-field effects. The first useable range gate covers the range 20 to 40 m and the maximum measurement distance was $d_{max}$ = 1280 m for EP1 on 2019-03-04 (Table 1). The half power beam width of the MRR is 1.5° resulting in a beam expansion of about 1.3 m at 100 m. The Nyquist velocity range is inverse proportional to the number of range gates $N$ (MRR Pro Manual, 2019) and was at the minimum for the first period with $v_{ny}$ = 24 m s⁻¹. The velocity resolution $\delta v$ of the MRR radial velocity $v_{MRR}$ is given by $v_{ny}$ /$m$. Because the wind direction was expected to vary depending on the general weather situation with snow potentially being blown either away or towards the MRR, the available velocity range $v_{ny}$ was set symmetrically to zero, resulting in an actual velocity range $v_{act} = \pm v_{ny}$ / 2 (Table 1). Velocities of $|v_{MRR}| > |v_{act}|$ result in aliasing (Tridon et al. 2011) but can be corrected for by applying a dealiasing procedure based on $v_{dealiased} = v_{MRR} + n\, v_{ny}$, where $n$ is the dealiasing number (integer with -1 if the lower limit of the Nyquist interval is exceeded and +1 if the upper limit is exceeded). However, particle velocities $|v_{MRR}| > |v_{act}|$ were rare. Another possible source of uncertainty of the Doppler velocity is the effect of ground clutter at small range gates, where the beam is not properly formed. However, since the MRR was installed at the edge of a steep slope (30°, Fig. 1b), the effects of ground echoes on the measured Doppler velocity can be neglected.

155  Furthermore, it is difficult to quantify an uncertainty on the mean Doppler velocity $v_{MRR}$ that is a moment of a distribution, the Doppler spectrum. The measure of the Doppler velocity itself is relatively precise, i.e. depends on the precision of the clock in the radar. It is more uncertain to which extent the mean Doppler velocity is representative of the movement of the particles within a range gate. However, the main wind direction was typically well aligned with the MRR view direction and the velocity fluctuations induced by turbulence is

160 assumed being normally distributed around the mean so that the mean Doppler velocity $v_{MRR}$ well represents the mean wind or particle velocity within a range gate. The averaging time was set to $T_i = 5$ s for EP1 and $T_i = 10$ s for EP2 and EP3.

Providing a recommendation for an ideal MRR parameter combination is difficult, as it depends on the transport distance and velocity of the blowing snow events. Based on the results of this study we recommend to start with a number of ($N = 32$) short ($\delta r = 10$ m) range gates resulting in a high distance resolution, a typically sufficient maximum measurement distance of 320

165 m and in a high Nyquist frequency of $v_{ny} = 48$ m s$^{-1}$ ($v_{act} = \pm 24$ m s$^{-1}$). A maximum possible value of $m = 256$ for the number of lines in spectrum results in a high velocity resolution of $\delta v = 0.19$ m s$^{-1}$. An averaging time of $T_i = 5$s seems to result in a sufficient temporal resolution without producing too much data while still capturing the major flow variability.

**Table 1: MRR parameter settings (Parameters 1. - 5.) for the three different evaluation periods investigated and the resulting**

170 **MRR limits (Parameters 6. - 9.):**

[revised manuscript text omitted]

**3.1 The Radar Reflectivity**

The radar reflectivity $Z$ is proportional to the fourth power of the diameter for snow particles (Ryzhkov 2019 and is thus mainly affected by the snow particle size and less so by the concentration as discussed before. The low reflectivity values of the measured pure blowing snow clouds (Fig. 3a), compared to the higher reflectivity of precipitation snowflakes (Fig. 3b), implies that the measured blowing snow clouds were composed of rather small particles. This is consistent with other findings of drifting and blowing snow investigations where small particle sizes of typically $50 - 500\,\mu m$ were detected (Nishimura and Nemoto 2005, Gromke et al. 2014) compared to precipitation snowflakes that can have diameters of several millimetres (e.g. Gergely et al. 2017). The lower reflectivities closer to the ridge ($d = 0 - 200$ m) compared to further away ($d > 300$ m) for the precipitation event (Fig. 3b) indicates smaller blowing snow particles due to higher wind speeds near the mountain ridge, whereas further away larger precipitation particles potentially dominate the backscatter of the radar signal.

[Figure]

240

**Figure 3:**  MRR reflectivity **for a**  **part of EP2 (2019-03-06 – 07) for pure** blowing snow events and **b)** EP3 (2019-03-14) **for blowing snow with concurrent snow precipitation.**

245 **3.****2 Radial Velocity and Turbulence Intensity: Exemplary cases**

The MRR radial velocity $v_{MRR}$ (Eq. 2) within a range gate is computed as the average of the MRR Doppler spectrum (MRR Pro Manual 2016) and is directly related to the blowing snow particle cloud velocity in the viewing direction of the MRR. In this Section we introduce the basic MRR data by means of four exemplary blowing snow events (Fig. 4) including a brief discussion and interpretation of the results as this data forms the basis for the analyses presented in the following Sections.

250   Fig. 4a shows the MRR radial velocity $v_{MRR}$ of the four blowing snow events of different characteristics within a two-minute time frame during EP1. The first event (No. 1) lasted for 25 s with a constant transport distance of 60 m. For the subsequent range gates (> 60 m), no snow particles were in the field of view of the MRR anymore (Fig. 1b). The assumption is that the snow was blown off the ridge horizontally by up to about 60 m before it started settling, either resulting in local accumulation or being further advected closer to the ground, and thus leaving the field of view of the MRR. Event No.

255   1 started with relatively high MRR radial velocities of about $\underline{v_{MRR} =}$ 10 - 11 m s$^{-1}$, while the velocities gradually decreased to about $\underline{v_{MRR} =}$ 7-8 m s$^{-1}$ towards the end of this event. The Sonic wind velocities (Fig. 4c) are in good agreement, also decreasing to about $v_{Sonic}$ = 8 m s$^{-1}$ towards the end of

$$\underline{I_{MRR} = \frac{\sigma_{v,MRR}}{v_{MRR}},} \qquad\qquad\qquad \text{(4)}$$

260   event No. 1. The turbulence intensity $I_{MRR}$ = 0.06 - 0.12 of this first event (Fig. 4b) shows low velocity fluctuations of the particle cloud, indicating a rather stable, low-level low-turbulence jet, which is supported by the Sonic turbulence intensities (Fig. 4d). The velocity drop at the end of event No. 1 is likely the reason for the break in snow being blown off the ridge between event No. 1 and 2.

[Figure]

**Figure 4: a) MRR radial velocity in the azimuth direction 22° for a two-minute period containing four different blowing snow events on 2019-03-04. b) Corresponding turbulence intensity *I*, Sonic c) wind velocity (absolute and in the direction 202°) and d) turbulence intensity for 5 s intervals.**

Blowing snow event No. 2 is different, starting with lower radial velocities of about $v_{MRR} =$ 9 m s$^{-1}$, likely being initiated by again higher wind velocities starting around 04:16:00 (Fig. 4c), then suddenly dropping to about $v_{MRR} =$ 6-7 m s$^{-1}$ during the following 10 s, because of another wind velocity $v_{Sonic}$ decrease around 04:16:10 (Fig. 4c). Strong velocity changes are an indication for turbulent gusts which is supported by higher MRR turbulence intensities of up to $I_{MRR} = 0.27$ (Fig. 3b4b). The

maximum turbulence intensity at the SDS measured with the Sonic in the direction of the MRR during event

275 No. 2 was $I_{Sonic}$ = 0.25 (Fig. 4d), thus in good agreement with the MRR result. However, the temporal agreement of the peak turbulence intensity is rather poor, as the peak in $I_{Sonic}$ lags the peak in $I_{MRR}$ although it should be vice versa. Nevertheless, an overall good agreement between the turbulence intensities measured with the Sonic and that of the first range gate of the MRR is found, with a mean difference of $\Delta I = mean(I_{MRR}$ - $I_{Sonic}$) = 0.01 with standard deviation of $\sigma_{\Delta I}$ = 0.09 for the entire EP1 and EP2. The lower velocity particle cloud of

280 event No. 2 is transported further within the field of view of the MRR compared to event No. 1, resulting in a gradually increasing transport distance starting from 60 m, increasing to 80 m, 120 m and finally to 140 m after 20 s. Interestingly, $v_{MRR}$ is increasing with distance for event No. 2, which is counter-intuitive, as one would rather expect a decrease of the wind velocity behind the ridge. However, the highly turbulent flow with changes in the wind direction and potentially large eddies of up to 100 m is likely causing this effect of higher velocities at longer distances. Events No. 3 and 4 both show rather high

285 radial velocities similarly to event No. 1, which are in good agreement with the Sonic wind velocities (Fig. 4c), but with slightly higher turbulence intensities indicating a more turbulent flow unlike for event No. 1. The transport distances are about 80 - 100 m for event No. 3 and 4.

Based on the above discussion of the four blowing snow events it seems that stronger turbulent fluctuations with higher turbulence intensities result in longer transport distances. This leads us to the hypothesis that not necessarily low-

290 turbulence jets with high wind velocities but turbulent gusts with lower wind velocities may be more effective in transporting blowing snow over longer distances on the lee side of a mountain ridge. Another explanation could be that the blowing snow cloud is vertically more extended for turbulent gusts which increases the likelihood of snow particles being in the field of view of the MRR (Fig. 1b), whereas for low-level low-turbulence jets the particles may rather quickly settle after a certain distance, leaving the field of view of the MRR. These considerations are further discussed in Section 3.4.

295 **3.3 Blowing Snow Distances**

The MRR blowing snow distances $d$ for EP1 are shown in Fig. 5a. Typically, a minimum distance of about 60 m is reached whereas longer distances > 100 m appear rather seldom. The distances $d$ and particle cloud radial velocities $v_{MRR}$ (Fig. 5b) may be smaller than the real absolute distances and velocities, as blowing snow from various angles (Fig. 5c), not only straight in the view direction of the MRR were detected as mentioned earlier. Nevertheless, the main

300 wind direction was typically in overall good agreement with the view direction (202°) of the MRR (Fig. 5c), and the main interest of this

[Figure]

**Figure 45: a) Temporal evolution of the horizontal transport distance of all blowing snow events of EP1 (2019-03-04, 0400 UTC+1 – 1000 UTC+1). b) Wind velocity parallel to the MRR direction (202°) measured with the Sonic compared to the close range (20 - 40m) blowing snow radial velocities measured with the MRR (see Fig. 4a). c) Wind direction (mainly 180° - 220°) and d) momentum flux –u'w' calculated using the Sonic data.**

study is in snow being blown off perpendicular to the Gotschnagrat mountain ridge. A comparison between the MRR radial velocities $v_{MRR}$ of the first useable range gate ($d$ = 40 m) and the horizontal wind velocity $v_{Sonic}$ measured with the Sonic, both for the direction of 202°, is provided in Fig. 5b. A qualitatively good agreement is found despite some outliers. Very low MRR velocities around $v_{MRR}$ = 2.5 m s$^{-1}$ are either an instrument artefact because of very low blowing snow particle concentrations, or wind directions temporarily deviating significantly from the MRR field of view direction. Discrepancies between the MRR and the Sonic velocities may be the result of the spatial average distance of about 30 m between the first usable range gate $d$ = 40 m (with a measurement volume extending from 20 to 40 m) and the location of the Sonic in combination with the slightly varying wind direction. To assess a potential dependency of the velocity difference on the wind direction, Fig. 6 shows the relative difference between the MRR and the Sonic velocity as a function of the wind direction $\alpha$ for EP1-

EP3. A positive trend is found with a bias of $v_{MRR} > v_{Sonic}$ for wind directions $\alpha > \text{202}180°$. Nevertheless, an overall good
agreement between the MRR radial and Sonic velocity is found, with a mean difference of $\text{Δv} = mean(((v_{MRR}$
$- v_{Sonic}) / v_{Sonic}) = \text{0.48 m s}^{+}10\%$ and a standard deviation of $\sigma_{\text{Δv}} = \text{1.22 m s}^{+}, \pm 20\%$. The intersection of the linear fit with the
$v_{MRR} - v_{Sonic} = 0$ line for $\alpha = \text{167}170°$ (Fig. $\text{5}6$) suggests a stable wind direction in the vicinity of the MRR and the SDS for
winds coming from that direction. This result is most likely strongly related to the local topography (Fig. 2b) influencing the
nearby wind field and direction, where the mountain station is located west and another SW-NE oriented mountain ridge east
of the MRR and the SDS, resulting in a rather undisturbed flow for southerly winds .

[Figure]

[Figure]

**Figure 6: Relative difference** between MRR and Sonic wind velocity in the direction 202° as a function
of wind direction for EP1-EP3.

Fig. 4d5d shows the momentum flux $-u`w`$ calculated from the Sonic wind velocities, which is generally positive for EP1, indicating a downward momentum flux and an increase in wind velocity with height above the location of the Sonic. However, between 0615 UTC+1 and 0700 UTC+1, the momentum flux was negative, indicating a decreasing wind velocity with height above the Sonic and the presence of a low-level jet close to the ground constantly blowing from a direction of 180° (South). During this time period, the wind velocity was highest with up to 12 - 13 m s$^{-1}$ and long blowing snow distances were reached of typically > 80 m (Fig. 4a5a). Furthermore, the best agreement between the Sonic wind velocity and the MRR radial velocity was found for this period of stable wind conditions.

Very similar results were found for EP2 (Fig. 7). Longer transport distances (Fig. 7a) were typically obtained as a result of the higher wind velocities (Fig. 7b). The wind direction (Fig. 7c) was typically quite stable although there were two periods (2100 – 2200 UTC+1 and 2300 - 2330 UTC+1) where the wind direction varied significantly. The momentum flux (Fig. 7d) was negative in about 50% of the time, indicating a higher presence of low-level jets close to the ground compared to EP1.

[Figure]

**Figure 7: a) Temporal evolution of the horizontal transport distance of all blowing snow events of EP2 (2019-03-06 1800 UTC+1 – 2019-03-07 0200 UTC+1). b) Wind velocity parallel to the MRR direction (202°) measured with a Sonic compared to the close range (40 - 80m) blowing snow radial velocities measured with the MRR. c) Wind direction (mainly 180° - 220°) and d) momentum flux – u'w' calculated using the Sonic data.**

**3.4 Blowing Snow Statistics**

The relative frequency of occurrence of blowing snow transport distances from Fig. 5a are shown in Fig. 8a for EP1. In 80% of the time, no blowing snow was present or detected by the MRR (transport distance $d = 0$ m). No events were detected for a distance $d = 20$ m since this range gate cannot be used as discussed earlier. Only  few events were detected for a transport distance $d = 40$ m, although this range gate delivered continuous information on radial velocities for higher transport distances $d > 40$ m (Fig. 4a). Therefore, we expect that also for $d = 20$ m, only very few or no

360 events would have been detected by the MRR, resulting in a gap in the frequency distribution for $0 < d < 60$ m in Fig. 8a. We hypothesize that, if the wind is strong enough and above a threshold wind speed to entrain and transport snow in suspension, a minimum transport distance of $d = 60$ m is reached, which occurred for about 10% of the total time of observation for EP1 (including the ´no blowing snow´ time). For distances $d > 60$ m, the relative frequency decreases exponentially with an only once observed maximum distance of $d = 200$ m. The mean Sonic wind velocity

365 was 7.3 m s$^{-1}$ during EP1, which is only 6h long but sampled at a temporal resolution of 5 s resulting in 4320 samples, thus providing a good data basis for statistics.

[Figure]

370 **Figure 8: Histogram of the transport distance of all blowing snow events for  a) EP1 (Fig. 5a), and b) EP2 (Fig. 7a), including  exponential fits for distances larger than the minimal transport distance.**

The relative frequency of occurrence of blowing snow distances for EP2 (2019-03-06 1800 UTC+1 – 2019-03-07 0200 UTC+1) is shown in Fig. 8b. The mean wind velocity of 9.1 m s$^{-1}$ during these 8h (10s sampling) measured

375 with the Sonic was significantly higher compared to EP1 (7.3 m s$^{-1}$), resulting in a larger gap before the

minimal transport distance and higher overall transport distances of up to maximum $d = 280$ m. The higher minimal transport distance of $d = 120$ m compared to EP1 might be the result of stronger gusts during the more powerful storm of EP2 and the snow surface conditions and its erodibility. Despite some differences between the two distributions in Fig. 8, both show very similar characteristics with a gap before a minimal distance is reached and an exponential

380   decay afterwards. Therefore, those distributions seem to be generally valid providing a good representation of the frequency of blowing snow distances for mountain ridges. A dependency of the minimal transport distance and the frequency distribution on the strength of the storm event and snow cover conditions could be investigated in future more detailed studies.

To estimate a threshold wind velocity (e.g. Li and Pomeroy 1997) and thus the erodibility of the surrounding snow surface, boxplots of the Sonic wind velocity as a function of the

385   transport distance are provided in Fig. 9. The median wind velocity increases by about $\cancel{6}2$ m s$^{-1}$ for a transport distances increasing from $d = 40$ – 200 m for EP1 and about 5 m s$^{-1}$ ($d = \underline{80 – 280}$ m) for EP2. An extrapolation of the  wind velocity to $d = 0$  m provides an estimate of a  threshold  velocity of 7.5 m s$^{-1}$ for EP1 and 8.8 m s$^{-1}$ for EP2, a result that is in overall good agreement with other studies (e.g. Li and Pomeroy 1997). Note: The wind velocity threshold definition for particle transport used in this study, defined for

390   a height of 1.5 m (Sonic), is similar to that used in Li and Pomeroy (1997), who defined a threshold wind speed at 10 m above ground. These definitions are different to the traditional definition of a threshold friction velocity for particle entrainment and saltation (e.g. Schmidt 1980, Guyomarc'h and Mérindol 1998, Clifton et al. 2006 Walter et al. 2012). The fact that the estimated threshold for EP2 (Fig. 9b) is 1.3 m s$^{-1}$ higher than for EP1 (Fig. 9a) supports our previous hypothesis of different snow surface conditions with a reduced erodibility for EP2.

395

[Figure]

[Figure]

**Figure 9: Sonic  wind velocity as a function of the transport distance of the blowing snow events for a) EP1and b) EP2.**

Turbulent gusts at rather low velocities were found being potentially responsible for longer transport distances as discussed in Section 3.2 (Fig. 4a). To investigate whether these events or low-level low-turbulence jets with high wind velocities are more effective in transporting snow over long distances across a mountain ridge, the turbulence intensities of the last range gate defining the blowing snow transport distance (Fig. 4b) are plotted as a function of the transport distance (box plot) in Fig. 10. For  EP1(Fig. 10a) and distances $d \geq 80$ m, the median, the upper and lower quartiles, the whiskers and the outliers all show a decreasing trend with increasing distance, indicating that low-level low-turbulence jets with high wind velocities are more effective than highly turbulent gusts in transporting blowing snow over long distances across a mountain ridge for EP1. Nevertheless, as mentioned earlier, highly turbulent motions still may result in a higher vertical extension of blowing snow clouds and thus in an increased likelihood of being within the field of view of the MRR (Fig. 1b) for long distances. For the stronger storm event of EP2, the turbulence level was significantly higher with median intensities of $0.1 - 0.2$ ((< 0.5 for period one) (Fig. 10b), supporting the latter assumption. Strong low-turbulence jets may also result in a slight downward air flow right after the ridge and the blowing snow may quickly settle getting out of the field of view of the MRR. The turbulence statistics shown in Fig. 10 do thus not allow to draw a conclusion on whether low-level low-turbulence jets or turbulent gusts are more effective in transporting blowing snow over longer distances. However, highly turbulent flows are more likely to bring particles to greater heights and thus influence cloud processes. Measurements with a two MRR system oriented parallel at different heights could provide a conclusion on which of the two events is more effective in transporting snow over longer distances across  mountain ridges.

[Figure]

**Figure 8̶10:** Turbulence intensity determined from the MRR spectral width of the Doppler spectrum of the range gate defining the blowing snow transport distance (Fig. 3̶4a) as a function of transport distance for e̶v̶a̶l̶u̶a̶t̶i̶o̶n̶ ̶p̶e̶r̶i̶o̶d̶ a) o̶n̶e̶EP1, and b) t̶w̶o̶EP2.

420

**3.5 Snow Height Distribution**

To provide a first connection between mountain ridge blowing snow events and a subsequent snow height distribution in the vicinity, the measured snow height distribution (Fig. 2a) is discussed in the context of prevailing precipitation and wind conditions and related to the analysed blowing snow events in this Section. The spatial variation in snow height difference between 2019-03-29 and 2019-03-12 of the investigated area around the MRR (Fig. 2a) shows distinct patterns as a result of pre- and post-depositional accumulation and erosion processes. Strongly dark blue and dark red spotted areas of maximum snow depth differences are an artefact from wind gusts affecting the drone flights on 2019-03-12, resulting in erroneous photogrammetry measurements (see Methods Section 2). Nevertheless, the smooth areas of the snow depth map show that significant snow deposition occurred north of the SDS in between the two drone flights, while other regions were eroded.

The increased snow accumulation north of the MRR shown in Fig. 2a is the result of a combination of preferential deposition and blowing snow, i.e. pre- and post-depositional accumulation processes. Although the pure blowing snow events analyzed in the previous sub-sections took place about a week prior to this long-term observational period between the two drone flights, two major snow storm events were found being responsible for the accumulation during the 17 days between the two  flights on 2019-03-12 and 2019-03-29. Fig. 11a shows a comparison of the Sonic wind velocity and the MRR radial velocity (similar as in Fig. 5b and 7b) for the first precipitation event on 2019-03-14 EP3). For this precipitation event, the MRR particle velocities are also in good agreement with the Sonic wind velocity at similar levels of up to 8 m s$^{-1}$ as for the pure blowing snow events of EP1 and EP2. The wind direction was also well aligned with the MRR view axis and quite stable from S to SW (approx. 200°) for the entire  storm (Fig. 11b). We assume that the wind resulted in both, preferential deposition during the precipitation event but also in snow on the ground being entrained and transported during strong gusts from the ridge to the accumulation zone (Fig. 1b, 2a). This simultaneous appearance of pre- and post-depositional accumulation processes also occurred during the second snow storm on 2019-03-15, which was very similar but is not presented here. The wind rose shown in Fig. 2a summarizes the wind directions for wind velocities > 6 m s$^{-1}$, thus potentially blowing snow effective, for the 9 days period 2019-03-12 to 2019-03-21. On the latter day, the MRR and the instruments of the SDS were dismantled. However, although the wind rose does not cover the entire period between the two drone flights, it clearly shows that the blowing snow effective wind direction was stable from S to SW at least for the first half of the time between the two drone flights. Similar transport distances for the blowing snow events with concurrent precipitation (EP3) as for those without (EP1 and EP2) are assumed, based on the similarity of the wind direction and wind velocity. Therefore, the increased accumulation north of the ridge up to distances of 200 m (Fig. 2a) are very likely the result of the two blowing snow events with concurrent precipitation between the two drone flights. Although the wind velocities for EP3 (Fig. 11a) are slightly smaller than for EP1 and EP2, probably resulting in smaller transport distances than shown in Fig. 5a and 7a, the snow gets likely being transported further closer to the ground outside the field of view of the MRR before it is finally deposited, which

might explain increased accumulation for distances of up to $d = 220$ m (Fig. 2a). Although the local topography and the near ground wind velocities north of the ridge also influenced the small scale (meters) snow height distribution on the ground, the main conclusion is that an overall good agreement is found between the blowing snow direction, wind velocities, blowing
460    snow distances and the larger scale (several tens of meters) snow accumulation pattern.

[Figure]

[Figure]

**Figure 9̶11:** Precipitation event (EP3) **on 2019-03-14 with strong wind from the south resulting in blowing snow and preferential deposition north of the Snow Drift Station as shown in Fig. 2̶2a. a) Sonic  wind velocity and MRR radial velocity,  b) wind direction and c) momentum flux –u'w' calculated using the Sonic data (similar as in Fig. 4̶b5 and c̶)7).**

[Figure]

**4 Summary and Conclusions**

Our results show that radar measurements of blowing snow may deliver valuable information to improve our understanding of pre- and post-depositional snow accumulation or redistribution processes on larger scales. The Micro Rain Radar (MRR)
475 instrument provides characteristics of and statistics on blowing snow distances, its frequency of occurrence, particle cloud velocities and turbulence intensities. We found good agreement between the MRR blowing snow velocity and the Sonic wind velocity, and that a minimal horizontal blowing snow transport distance of 60 - 120 m is reached in the lee of a mountain ridge, depending on the strength of the storm event. The relative frequency of occurrence decreases exponentially for distances longer than the minimal transport distance, with a measured maximum distance of 280 m in our
480 case. It was not possible to draw a conclusion on whether low-level low-turbulence jets or turbulent gusts are more effective in transporting blowing snow over longer distances in the lee of a mountain ridge. The increased snow height distribution north of the measurement location (Fig. 2a) was found being the result of a combination of preferential deposition and blowing snow accumulation during at least two measured and analyzed snowstorm events. The presented snow height distributions together with the characterization of the blowing snow events provides a valuable data basis for validating coupled numerical
485 weather and snowpack simulations.

Further investigations are required for more clarification and may incorporate measurements with a second MRR system oriented parallel at a slightly different elevation to better resolve the local wind field and blowing snow events; particularly to capture the process of settling snow disappearing from the field of view of the upper MRR. The MRR instrument was also recently tested by the CRYOS group at EPFL Lausanne, Switzerland, for measuring vertical blowing snow velocity profiles
490 and its temporal variability in eastern Antarctica at the site S17 near the Japanese research station Syowa (unpublished work in progress), where blowing snow layers can reach a vertical extend of up to 200 m (Palm et al. 2017). The next challenge for radar specialists will be finding a way to extract particle concentrations from the radar measurements to estimate particle mass fluxes or at least its order of magnitude. Exploring the potential of horizontally pointing cloud physics LIDAR (e.g. Mona et al. 2012) in detecting the spatio-temporal variability of blowing snow would be worthwhile for the community
495 interested in characterizing and better understanding pre- and post-depositional snow accumulation processes in various cold regions worldwide.

**Author contribution**

BW, HH and ML designed the experiments and BW carried them out. JG provided MRR support and YB conducted the drone flights. BW prepared the manuscript with contributions from all co-authors.

**Data availability**

The MRR and SDS data for the entire RACLETS campaign will soon be available on the ENVIDAT data repository, but are available on request by then.

**Acknowledgments**

We would like to thank the SLF workshop for supporting us with the design and construction of the Snow Drift Station. Furthermore, we would like to thank the Environmental Remote Sensing Laboratory (LTE) at EPFL, especially Alexis Berne and Alfonso Ferrone for lending the MRR and for the technical support with the instrument. Thanks also to Andreas Stoffel, Elisabeth Hafner and Lucie Eberhard for performing the photogrammetric drone flights, David Wagner, Felix von Rütte and Beat Nett for their support with the installation of the snow drift station, and Rebecca Mott for the GIS support. Michael Lehning and Josué Gehring acknowledge the financial support from the Swiss National Science Foundation (grant 200020--179130  and 200020-175700/1).